# A General Framework For Proving The Equivariant Strong Lottery Ticket Hypothesis

**Damien Ferbach**[*]
École Normale Supérieure, PSL, and Mila [†]
`damien.ferbach@ens.psl.eu`

**Christos Tsirigotis** [*]
Université de Montréal and Mila

**Gauthier Gidel** [‡]
Université de Montréal and Mila

**Avishek (Joey) Bose**
McGill University and Mila

## Abstract

The Strong Lottery Ticket Hypothesis (SLTH) stipulates the existence of a subnetwork within a sufficiently overparameterized (dense) neural network that—when initialized randomly and without any training—achieves the accuracy of a fully trained target network. Recent works by da Cunha et al. (2022b); Burkholz (2022a) demonstrate that the SLTH can be extended to translation equivariant networks—i.e. CNNs—with the same level of overparametrization as needed for the SLTs in dense networks. However, modern neural networks are capable of incorporating more than just translation symmetry, and developing general equivariant architectures such as rotation and permutation has been a powerful design principle. In this paper, we generalize the SLTH to functions that preserve the action of the group $G$—i.e. $G$-equivariant network—and prove, with high probability, that one can approximate any $G$-equivariant network of fixed width and depth by pruning a randomly initialized overparametrized $G$-equivariant network to a $G$-equivariant subnetwork. We further prove that our prescribed overparametrization scheme is optimal and provides a lower bound on the number of effective parameters as a function of the error tolerance. We develop our theory for a large range of groups, including subgroups of the Euclidean $E(2)$ and Symmetric group $G \leq S_n$—allowing us to find SLTs for MLPs, CNNs, $E(2)$-steerable CNNs, and permutation equivariant networks as specific instantiations of our unified framework. Empirically, we verify our theory by pruning overparametrized $E(2)$-steerable CNNs, $k$-order GNNs, and message passing GNNs to match the performance of trained target networks.

## 1 Introduction

Many problems in deep learning benefit from massive amounts of annotated data and compute that enables the training of models with an excess of a billion parameters. Despite this appeal of overparametrization many real-world applications are resource-constrained (e.g., on device) and demand a reduced computational footprint for both training and deployment (Deng et al., 2020). A natural question that arises in these settings is then: is it possible to marry the benefits of large models—empirically beneficial for effective training—to the computational efficiencies of smaller sparse models?

A standard line of work for building *compressed* models from larger fully trained networks with minimal loss in accuracy is via weight pruning (Blalock et al., 2020). There is, however, increasing empirical evidence to suggest weight pruning can occur significantly prior to full model convergence. Frankle and Carbin (2019) postulate the extreme scenario termed *lottery ticket hypothesis* (LTH) where a subnetwork extracted at initialization can be trained to the accuracy of the parent network—in effect "winning" the weight initialization lottery. In an even more striking phenomenon Ramanujan et al. (2020) find that not only do such sparse subnetworks exist at initialization but they already achieve impressive performance without any training. This remarkable occurrence termed the

---

[*]denotes equal contribution.

[†]Work done during an internship at Mila

[‡]Canada Cifar AI Chair

*strong lottery ticket hypothesis* (SLTH) was proven for overparametrized dense networks with no biases (Malach et al., 2020; Pensia et al., 2020; Orseau et al., 2020), non-zero biases (Fischer and Burkholz, 2021), and vanilla CNNs (da Cunha et al., 2022b). Recently, Burkholz (2022b) extended the work of Pensia et al. (2020) to most activation functions that behave like ReLU around the origin, and adopted another overparametrization framework as in Pensia et al. (2020) such that the overparametrized network has depth $L + 1$ (no longer $2L$). However, the optimality with respect to the number of parameters (Theorem 2 in Pensia et al. (2020)) is lost with this method. Moreover, Burkholz (2022a) extended the results of da Cunha et al. (2022b) on CNNs to non-positive inputs.

Modern architectures, however, are more than just MLPs and CNNs and many encode data-dependent inductive biases in the form of equivariances and invariances that are pivotal to learning smaller and more efficient networks (He et al., 2021). This raises the important question: can we simultaneously get the benefits of equivariance and pruning? In other words, does there exist winning tickets for the equivariant strong lottery for general equivariant networks given sufficient overparametrization?

**Present Work**. In this paper, we develop a unifying framework to study and prove the existence of strong lottery tickets (SLTs) for general equivariant networks. Specifically, in our main result (Thm. 1) we prove that any fixed width and depth target $G$-equivariant network that uses a point-wise ReLU can be approximated with high probability to a pre-specified tolerance by a subnetwork within a random $G$-equivariant network that is overparametrized by doubling the depth and increasing the width by a logarithmic factor. Such a theorem allows us to immediately recover the results of Pensia et al. (2020); Orseau et al. (2020) for MLPs and of Burkholz et al. (2022); da Cunha et al. (2022b) for CNNs as specific instantiations under our unified equivariant framework. Furthermore, we prove that a logarithmic overparametrization is necessarily optimal—by providing a lower bound in Thm. 2—as a function of the tolerance. Crucially, this is *irrespective* of which overparametrization strategy is employed which demonstrates the optimality of Theorem 1. Notably, the extracted subnetwork is also $G$-equivariant, preserving the desirable inductive biases of the target model; such a fact is importantly not achievable via a simple application of previous results found in (Pensia et al., 2020; da Cunha et al., 2022b).

Our theory is broadly applicable to any equivariant network that uses a pointwise ReLU nonlinearity. This includes the popular E(2)-steerable CNNs with regular representations (Weiler and Cesa, 2019) (Corollary 1) that model symmetries of the $2d$-plane as well as subgroups of the symmetric group of $n$ elements $\mathcal{S}_n$, allowing us to find SLTs for permutation equivariant networks (Corollary 2) as a specific instantiation. We substantiate our theory by conducting experiments by explicitly computing the pruning masks for randomly initialized overparametrized E(2)-steerable networks, $k$-order GNNs, and MPGNNs to approximate another fully trained target equivariant network.

## 2 BACKGROUND AND RELATED WORK

**Notation and Convention**. For $p \in \mathbb{N}$, $[p]$ denotes $\{0, \cdots, p-1\}$. We assume that the starting index of tensors (vectors, matrices,...) is 0, e.g., $\mathbf{W}_{p,q}$, $p, q \in [d]$. $G$ is a group, and $\rho$ is its representation. We use $|\cdot|$ for the cardinality of a set, while $\bigoplus$ represents the direct sum of vector spaces or group representations and $\otimes$ indicates the Kroenecker product. We use $*$ to denote a convolution. We define $x^+, x^-$ as $x^+ = \max(0, x)$ and $x^- = \min(0, x)$. $\|\cdot\|$ is a $\ell_p$ norm while $\|\|\cdot\|\|$ is its operator norm. For a basis $\mathcal{B} = \{b_1, \ldots, b_p\}$, we write $\|\|\mathcal{B}\|\| = \max_{\|\alpha\|_\infty \leq 1} \|\|\sum_{k=1}^p \alpha_k b_k\|\|$. $\sigma(x) = x^+$ is the pointwise ReLU. Finally, we take $(\epsilon, \delta) \in [0, \frac{1}{2}]^2$, and $\mathcal{U}([a, b])$ is the uniform distribution on $[a, b]$.

**Equivariance**. We are interested in building equivariant networks that encode the symmetries induced by a given group $G$ as inductive biases. To act using a group we require a group representation $\rho : G \to GL(\mathbb{R}^D)$, which itself is a group homomorphism and satisfies $\rho(g_1 g_2) = \rho(g_1)\rho(g_2)$ as $GL(\mathbb{R}^D)$ is the group of $D \times D$ invertible matrices with group operation being ordinary matrix multiplication. Let us now recall the main definition for equivariance:

**Definition 2.1.** *Let $\mathcal{X} \subset \mathbb{R}^{D_x}$ and $\mathcal{Y} \subset \mathbb{R}^{D_y}$ be two sets with an action of a group $G$. A map $f : \mathcal{X} \to \mathcal{Y}$ is called $G$-equivariant, if it respects the action, i.e., $\rho_\mathcal{Y}(g)f(x) = f(\rho_\mathcal{X}(g)x), \forall g \in G$ and $x \in \mathcal{X}$. A map $h : \mathcal{X} \to \mathcal{Y}$ is called $G$-invariant, if $h(x) = h(\rho_\mathcal{X}(g)x), \forall g \in G$ and $x \in \mathcal{X}$.*

As a composition of equivariant functions is equivariant, to build an equivariant network it is sufficient to take each layer $f_i$ to be $G$-equivariant and utilize a $G$-equivariant non-linearity (e.g. pointwise ReLU). Given a vector space and a corresponding group representation we can define a feature space $\mathbb{F}_i := (\mathbb{R}^{D_i}, \rho_i)$. Note that we can stack multiple such feature spaces in a layer, for example, the input feature space to an equivariant layer $i$ can be written as $n_i$ blocks $\mathbb{F}_i^{n_i} := \bigoplus_{m=1}^{n_i} \mathbb{F}_i$.

A $G$-equivariant basis is a basis of the space of equivariant linear maps between two vector spaces. We can decompose a $G$-equivariant linear map $f_i : \mathbb{F}_i \rightarrow \mathbb{F}_{i+1}$ in a corresponding equivariant basis $\mathcal{B}_{i \rightarrow i+1} = \{b_{i \rightarrow i+1,k} \in \mathbb{R}^{D_i \times D_{i+1}}, \forall k \in [|\mathcal{B}_{i \rightarrow i+1}|]\}$. When working with stacks of $n_i$ (resp. $n_{i+1}$) input (resp. output) feature spaces we may express the full equivariant basis by considering $\kappa_{n_i \rightarrow n_{i+1}} = \{\kappa^{p,q}_{n_i \rightarrow n_{i+1}} \in \mathbb{R}^{n_i \times n_{i+1}}, (p,q) \in [n_i] \times [n_{i+1}]\}$, where each element $\kappa^{p,q}_{n_i \rightarrow n_{i+1}}$ is a matrix with a single non-zero entry at position $(p,q)$. Then the basis for $G$-equivariant maps between $\mathbb{F}^{n_i}_i \rightarrow \mathbb{F}^{n_{i+1}}_{i+1}$ can be written succinctly as the Kronecker product between two basis elements $\kappa_{n_i \rightarrow n_{i+1}} \otimes \mathcal{B}_{i \rightarrow i+1}$. Some instances of $G$ and $\mathbb{F}_i$ are presented in Tab. 2. For example, in the case of CNNs with kernel size $d^2$, the linear map $f$ is a convolution where $n_i$ (resp. $n_{i+1}$) are the number of input (resp. output) channels and $\kappa_{n_i \rightarrow n_{i+1}} \otimes \mathcal{B}_{i \rightarrow i+1}$ is the basis of convolutions of size $d^2 \times n_i \times n_{i+1}$.

**Related Work on Strong Lottery Tickets**. Winning SLTs approximate a target ReLU network $f(x)$ by pruning an overparametrized ReLU network $g(x)$ with weights in any given layer drawn i.i.d. from $w_i \sim \mathcal{U}([-1,1])$.[1] Our error metric of choice is the uniform approximation over a unit ball: $\max_{x \in \mathbb{R}^D : ||x|| \leq 1} ||f(x) - \hat{g}(x)|| \leq \epsilon$, where $\hat{g}(x)$ is the subnetwork constructed from pruning $g(x)$. Let us first consider the case of approximating a single neuron $w_i \in [-1,1]$ in some layer of $f(x)$ with $n$ i.i.d. samples $X_1, \ldots, X_n \sim \mathcal{U}([-1,1])$. If $n = O(1/\epsilon)$ then there exists a $X_i$ that is $\epsilon$-close to $w_i$ (Malach et al., 2020). A similar approximation fidelity can be achieved with an exponentially smaller number of samples by not relying on just a single $X_i$ but instead a subset whose sum approximates the target weight. Lueker (1998); da Cunha et al. (2022a) proved that $n = O(\log(1/\epsilon))$ random variables were sufficient for the existence of a solution to the random SUBSET-SUM problem (a subset $S \subseteq \{1, \ldots, n\}$ such that $|w_i - \sum_{i \in S} X_i| \leq \epsilon$). Pensia et al. (2020) utilize the SUBSET-SUM approach for weights on dense networks resulting in a logarithmic overparametrization of the width of a layer in $g(x)$. To bypass the non-linearity (ReLU) Pensia et al. (2020) decompose the output activation $\sigma(wx) = w^+ x^+ + w^- x^-$ and approximate each term separately. With no additional assumption on the inputs (da Cunha et al. (2022b) assume positive entries), this approach fails for equivariant networks as each entry of the output of an equivariant linear map is affected by multiple input entries.

# 3  SLT FOR GENERAL EQUIVARIANT NETS

Our results and proof techniques build upon the line of work by Pensia et al. (2020), da Cunha et al. (2022b), and Burkholz (2022a). Specifically, we rely on the SUBSET-SUM algorithm (Lueker, 1998) to aid in approximating any given parameter of the target network. Departing from prior work, the main idea used in our technical analysis is to prune an overparametrized equivariant network in a way that *preserves equivariance*, as applying SUBSET-SUM using da Cunha et al. (2022b) construction may destroy the prunned network's equivariance.

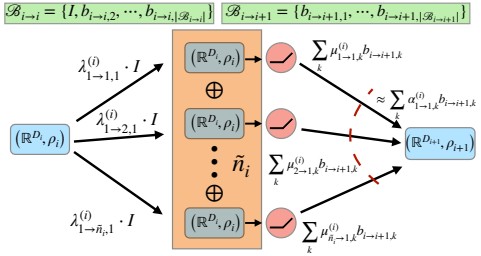

Figure 1: General Equivariant Pruning Method

**Challenges in Adapting Proof Techniques**. There are two major difficulties in adapting the tools first introduced in Pensia et al. (2020) to $G$-steerable networks. In proving the SLTH for dense networks the relevant parameters that can be pruned are all the parameters of weight matrices, which can be intuitively understood as pruning in a canonical basis. However, such a strategy immediately fails for $G$-equivariant maps as the canonical basis is not generally $G$-equivariant, thus pruning in this basis breaks the structure of the network and its equivariance. In fact, as described in Weiler and Cesa (2019) a $G$-equivariant linear map consists of linearly combining the elements of the equivariant basis with learned combination coefficients which are the effective parameters of the $G$-equivariant model. To preserve equivariance we may only prune these parameters and *not* any weight in $f_i$. However, this introduces a new complication as the interaction with the ReLU becomes more challenging. da Cunha et al. (2022b) circumvent this in the special case of regular CNNs by assuming only positive inputs. In contrast, our main technical lemma (Lem. 1), introduces a construction that does not require such a restrictive assumption and generalizes the techniques of Burkholz (2022a) to $G$-equivariant networks.

**Overparameterized Network Shape**. We seek to approximate a single $G$-equivariant layer with two random overparameterized $G$-equivariant layers. We take the input $||x|| \leq 1$ to be

---
[1]It will work with any distribution which contains a uniform distribution, e.g. Gaussian, see §A

in a bounded domain to control the error which could diverge on unbounded domains. Let $\mathcal{F}_i$ be the set of $G$-equivariant linear maps $\mathbb{F}_i^{n_i} \rightarrow \mathbb{F}_{i+1}^{n_{i+1}}$ of the $i$-th layer in the target network. Then, $f_i \in \mathcal{F}_i$ s.t., $\|\|f_i\|\| \leq 1$, is a specific realization of a target equivariant map that we will approximate—i.e. $f_i(x) = \mathbf{W}_i^f(x)$. Without any loss of generality, let the coefficients of $\mathbf{W}_i^f$ be such that $|\alpha_k| \leq 1$ when decomposed in the basis $\kappa_{n_i \rightarrow n_{i+1}} \otimes \mathcal{B}_{i \rightarrow i+1}$. Concretely, $f_i \in \mathcal{F}_i := \left\{ \mathbf{W}_i^f = \sum_k \alpha_k b_k : b_k \in \kappa_{n_i \rightarrow n_{i+1}} \otimes \mathcal{B}_{i \rightarrow i+1}, |\alpha_k| \leq 1, \|\|\mathbf{W}_i^f\|\| \leq 1 \right\}$. We can now recursively apply the previous constructions to construct a desired $G$-equivariant target network $f \in \mathcal{F}$ of depth $l \in \mathbb{N}$. Analagously, we can define an atomic unit of our random overparameterized source model $\mathcal{H}_i$ as the set of $G$-equivariant maps with one intermediate feature space (layer) $\mathbb{F}_i^{\tilde{n}_i}$ followed by a ReLU. That is, any $h_i \in \mathcal{H}_i$ applied to an input $x$ can be written as $h_i(x) = \mathbf{W}_{2i+1}^h \sigma(\mathbf{W}_{2i}^h x)$. In our construction, we choose $\mathbf{W}_{2i}^h$ whose equivariant basis is $\kappa_{n_i \rightarrow \tilde{n}_i} \otimes \mathcal{B}_{i \rightarrow i}$ where $\tilde{n}_i$ is the overparametrization factor of the $i$-th layer. We assume $\mathcal{B}_{i \rightarrow i}$ contains the identity element, which is trivially equivariant. The basis coefficients of $\mathbf{W}_{2i}^h$ are written as $\lambda_{p \rightarrow q, k}^{(i)}$, which refers to the coefficient of the $k$-th basis element in $\mathcal{B}_{i \rightarrow i}$ for the map between the $p$-th block of $\mathbb{F}_i^{n_i}$ to the $q$-th block of $\mathbb{F}_i^{\tilde{n}_i}$. Similarly, $\mathbf{W}_{2i+1}^h$ can be decomposed in the basis $\kappa_{\tilde{n}_i \rightarrow n_{i+1}} \otimes \mathcal{B}_{i \rightarrow i+1}$ with coefficients $\mu_{p \rightarrow q, k}^{(i)}$. Fig. 1 illustrates this construction after pruning the first layer for $n_i = n_{i+1} = 1$ which leads to a "diamond" shape. We can finally apply the previous construction to build an overparametrized network $h \in \mathcal{H}$ of depth $2l$. We summarize all the notation used in the rest of the paper in Tab. 1.

| $G$-Equivariant map | Basis | Basis Coefficients | | |
|---|---|---|---|---|
| $\mathbf{W}_{2i}^h : \mathbb{F}_i^{n_i} \rightarrow \mathbb{F}_i^{\tilde{n}_i}$ | $\kappa_{n_i \rightarrow \tilde{n}_i} \otimes \mathcal{B}_{i \rightarrow i}$ | $\lambda_{p \rightarrow q, k}^{(i)},$ | $p \in [n_i], q \in [\tilde{n}_i], k \in [|\mathcal{B}_{i \rightarrow i}|]$ | |
| $\mathbf{W}_{2i+1}^h : \mathbb{F}_i^{\tilde{n}_i} \rightarrow \mathbb{F}_{i+1}^{n_{i+1}}$ | $\kappa_{\tilde{n}_i \rightarrow n_{i+1}} \otimes \mathcal{B}_{i \rightarrow i+1}$ | $\mu_{p \rightarrow q, k}^{(i)},$ | $p \in [\tilde{n}_i], q \in [n_{i+1}], k \in [|\mathcal{B}_{i \rightarrow i+1}|]$ | |
| $\mathbf{W}_i^f : \mathbb{F}_i^{n_i} \rightarrow \mathbb{F}_{i+1}^{n_{i+1}}$ | $\kappa_{n_i \rightarrow n_{i+1}} \otimes \mathcal{B}_{i \rightarrow i+1}$ | $\alpha_{p \rightarrow q, k}^{(i)},$ | $p \in [n_i], q \in [n_{i+1}], k \in [|\mathcal{B}_{i \rightarrow i+1}|]$ | |

Table 1: Summary of notation used to decompose each $G$-equivariant map in the source and target networks.

## 3.1 THEORETICAL RESULTS

We first prove Lemma 1 which states that with high probability a random overparametrized $G$-equivariant network of depth $l = 2$ (Fig. 1) can $\epsilon$-approximate any target map in $\mathcal{F}_i$ via pruning.

**Lemma 1.** *Let $h_i \in \mathcal{H}_i$ be a random overparametrized $G$-equivariant network as defined above, with coefficients $\lambda_{p \rightarrow q, k}^{(i)}$ and $\mu_{p \rightarrow q, k}^{(i)}$ drawn from $\mathcal{U}([-1, 1])$. Further suppose that each $\tilde{n}_i = C_1 n_i \log(\frac{n_i n_{i+1} \max(|\mathcal{B}_{i \rightarrow i+1}|, \|\|\mathcal{B}_{i \rightarrow i+1}\|\|)}{\min(\epsilon, \delta)})$ where $C_1$ is a constant. Then, with probability $1 - \delta$, for every target $G$-equivariant layer $f_i \in \mathcal{F}_i$, one can find two pruning masks $\mathbf{S}_{2i}, \mathbf{S}_{2i+1}$ on the coefficients $\lambda_{p \rightarrow q, k}^{(i)}$ and $\mu_{p \rightarrow q, k}^{(i)}$ respectively such that:*

$$\max_{\mathbf{x} \in \mathbb{R}^{D_i \times n_i}, \|\mathbf{x}\| \leq 1} \|(\mathbf{S}_{2i+1} \odot \mathbf{W}_{2i+1}^h)\sigma((\mathbf{S}_{2i} \odot \mathbf{W}_{2i}^h)\mathbf{x}) - f_i(\mathbf{x})\| \leq \epsilon. \tag{1}$$

*Proof sketch.* We prune all non-identity coefficients of the basis decomposition of the first layer obtaining "diamond" shape (see Fig. 1 for $(n_i = n_{i+1} = 1)$) allowing us to bypass the pointwise ReLU. The two layers can now be used to approximate every weight of the target by solving independent SUBSET-SUM problems on the coefficients of the second layer. The full proof is provided in §B.1. $\square$

To approximate any $f$ in $\mathcal{F}_i$ which is a $G$-equivariant target network of depth $l$ and fixed width, we can now apply Lemma 1 $l$-times to obtain our main theorem, whose proof is provided in §B.2.

**Theorem 1.** *Let $h \in \mathcal{H}$ be a random overparametrized $G$-equivariant network with coefficients $\lambda_{p \rightarrow q, k}^{(i)}$ and $\mu_{p \rightarrow q, k}^{(i)}$, for $i \in [l]$ and indices $p, q, k$ as defined in Table 1, all drawn from $\mathcal{U}([-1, 1])$. Suppose that $\tilde{n}_i = C_2 n_i \log(\frac{n_i n_{i+1} \max(|\mathcal{B}_{i \rightarrow i+1}|, \|\|\mathcal{B}_{i \rightarrow i+1}\|\|)l}{\min(\epsilon, \delta)})$, where $C_2$ is a constant. Then with probability $1 - \delta$, for every $f \in \mathcal{F}$, one can find a collection of pruning*

*masks* $\mathbf{S}_{2l-1}, \dots \mathbf{S}_0$ *on the coefficients* $\lambda_{p \to q, k}^{(i)}$ *and* $\mu_{p \to q, k}^{(i)}$ *for every layer* $i \in [l]$ *such that:*

$$\max_{\mathbf{x} \in \mathbb{R}^{D_0 \times n_0}, \, \|\mathbf{x}\| \leq 1} \|(\mathbf{S}_{2l-1} \odot \mathbf{W}_{2l-1}^h) \sigma \left( \dots \sigma((\mathbf{S}_0 \odot \mathbf{W}_0^h) \mathbf{x}) \right) - f(\mathbf{x})\| \leq \epsilon. \quad (2)$$

We recover a similar overparametrization as Pensia et al. (2020) with respect to the width of $h$. However, the significant improvement provided by this result is that, since we do not prune dense nets but $G$-equivariant ones, the number of effective parameters in the overparametrized network is $|\mathcal{B}_{i \to i+1}|/D_i D_{i+1}$ smaller than a dense net of the same width. In section 3.2 we make this difference explicit and show Theorem 1 is optimal up to log factors not only with respect to the tolerance $\epsilon$ but also with respect to $|\mathcal{B}_{i \to i+1}|/D_i D_{i+1}$ quantifying the expressiveness of $G$-equivariant networks.

## 3.2 LOWER BOUND ON THE OVERPARAMETRIZATION

When searching for equivariant winning tickets a natural question that arises is the optimality of the overparametrization factor $\tilde{n}_i$ with respect to the tolerance $\epsilon$. In the same vein as Pensia et al. (2020) for MLPs, we now prove under mild assumptions that, in the equivariant setting, $\tilde{n}_i$ is indeed optimal (Theorem 2). We will assume that our equivariant basis $\mathcal{B}_{i \to i+1}$ has the following property: $\forall f_i \in \text{Span}(\mathcal{B}_{i \to i+1})$ where $f_i = \sum_k \alpha_k b_{i \to i+1, k}$ we have: $\|f_i\| \leq 1 \implies |\alpha_k| \leq 1, \, k \geq 0$. Note that this can be obtained by a rescaling of the basis elements. Lastly, we also assume the existence of positive constants $M_1$ and $M_2$ such that $|\mathcal{B}_{i \to i}| \leq M_1 |\mathcal{B}_{i \to i+1}|$ and $n_i \leq M_2 n_{i+1}$. These assumptions are relatively mild and hold in the practical situations described in Tab. 2 (cf §B.3 for details). Under these assumptions we achieve the following (tight) lower bound.

**Theorem 2.** *Let* $\hat{h}_i$ *be a network with* $\Theta$ *parameters such that:*

$$\forall f_i \in \mathcal{F}_i, \, \exists S_i \in \{0, 1\}^{\Theta} \text{ such that } \max_{\mathbf{x} \in \mathbb{R}^{D_i \times n_i}, \, \|\mathbf{x}\| \leq 1} \|(S_i \odot \hat{h}_i)(\mathbf{x}) - f_i(\mathbf{x})\| \leq \epsilon. \quad (3)$$

*Then* $\Theta$ *is at least* $\Omega \left( n_i n_{i+1} |\mathcal{B}_{i \to i+1}| \log(\frac{1}{\epsilon}) \right)$ *and* $\tilde{n}_i$ *is at least* $\Omega(n_i \log \left( \frac{1}{\epsilon} \right))$ *in Theorem 1.*

*Proof Idea.* The full proof is provided in §B.3 and relies on a counting argument to compare the number of pruning masks and functions in $\mathcal{F}_i$ within a distance of at least $2\epsilon$ of each other. □

Thm. 2 dictates that if we wish to approximate a $G$-equivariant network target network to $\epsilon$-tolerance by pruning an overparametrized arbitrary network, the latter must have at least $\Omega(n_i n_{i+1} |\mathcal{B}_{i \to i+1}| \log(\frac{1}{\epsilon}))$ parameters. Applying the above result to our prescribed overparametrization scheme in Thm. 1 we find our proposed strategy is optimal with respect to $\epsilon$ and almost optimal with respect to $|\mathcal{B}_{i \to i+1}|$. We incur a small extra log factor whose origin is discussed in §B.3. In the equivariant setting, the result in Pensia et al. (2020) is far from optimal as their result gives guarantees on the pruning of dense nets with a similar width as the $G$-equivariant targets which incurs an increase by a factor $D_i D_{i+1}/|\mathcal{B}_{i \to i+1}|$ in the number of parameters. As a specific example, for overparametrized $G$-steerable networks (Tab. 2), we have $D_i D_{i+1}/|\mathcal{B}_{i \to i+1}| = d^2 |G|$. On images of shape $\mathbb{R}^{224 \times 224 \times 3}$ with $G = C_8$, it corresponds to $\approx 4.10^5$ fewer "effective" parameters than a dense network. Finally, we note that Thm. 2 makes no statement on which overparametrization strategy *achieves* such a lower bound. Remarkably, the pruning strategy prescribed by Thm. 1 recovers this optimal lower bound on $\tilde{n}_i$, meaning that, unsurprisingly, $G$-equivariant nets are the most suitable structure to prune.

## 4 SLT FOR SPECIFIC CHOICES OF $G$

In this section, we turn our focus to specific instantiations of our main theoretical results for different choices of groups. To apply Theorem 1, one simply needs to specify the group $G$, the group representation $\rho(g)$, and finally the feature space $\mathbb{F}$. For instance, we can immediately recover the results for dense networks (Pensia et al., 2020) by noticing $G = \{e\}$ is the trivial group with a trivial action on $\mathbb{R}^D$ (see the proof in §C). In Table 2 below we highlight different $G$-equivariant architectures through the framework provided in §3 before proving each setting in the remainder of the section.

### 4.1 A CASE STUDY WITH CNNS

As a warmup, let us consider the case of vanilla CNNs that possess translation symmetry. In this case, $G = (\mathbb{Z}^2, +)$ the group of translations of the plane and $D_i = d^2$ where $d^2$ is the size of a feature map

| | $G$ | $\rho_i$ | $\mathbb{F}_i$ | $\|\mathcal{B}_{i \to i+1}\| \vee \|\|\mathcal{B}_{i \to i+1}\|\|$ |
|---|---|---|---|---|
| MLP | $\{e\}$ | trivial | $\mathbb{R}$ | $1$ |
| CNN | $(\mathbb{Z}^2, +)$ | $f_i(x - t)$ | $(\mathbb{R}^{d^2}, \rho_i)$ | $d^2$ |
| E(2)-CNN | $(\mathbb{Z}^2, +) \rtimes \mathrm{O}(2)$ | $\rho_{\mathrm{reg}}(g) f_i(g^{-1}(x - t))$ | $(\mathbb{R}^{d^2 \times \|G\|^2}, \rho_i)$ | $d^2 \|G\|^3$ |
| Permutation | $\mathcal{S}(n)$ | $\mathbf{X}_{i_{\sigma(1)}, \dots, i_{\sigma(k)}, j}$ | $(\mathbb{R}^{n^{k_i}}, \rho_i)$ | $\tilde{b}(k_i + k_{i+1}) \vee (n^{k_i} + 1)$ |

Table 2: Instantiations of Theorem 1 for different choices of $G$. MLP was proven in Pensia et al. (2020), CNN was proven in da Cunha et al. (2022b); Burkholz (2022a). We note $a \vee b := \max(a, b)$.

at layer $i$. Finally, $\rho_i$ acts on the feature space $\mathbb{R}^{d^2}$ by translating the coordinates of a point in the plane. The equivariant basis of $f_i$ in this setting (what we denoted $\mathcal{B}_{i \to i+1}$ in the general case) are convolutions with kernels $\mathbf{K}_i^f \in \mathbb{R}^{d^2 \times n_i \times n_{i+1}}$ that are built using the canonical basis and $n_i$ and $n_{i+1}$ are the input/output channels. We can apply Thm. 1 to achieve Cor. 4 (see §D for details) which recovers Burkholz (2022a, Thm. 3.1) and is a strict generalization of the result by da Cunha et al. (2022b).

## 4.2 SLT FOR E(2) STEERABLE NETS

The Euclidean group E(2) is the group of isometries of the plane $\mathbb{R}^2$ and is defined as the semi-direct product between the translation and orthogonal groups of two dimensions $(\mathbb{R}^2, +) \rtimes \mathrm{O}(2)$ with elements $(t, g) \in \mathrm{E}(2)$ being shifts and planar rotations or flips. The most general method to build equivariant networks for E(2) is in the framework of steerable $G$-CNN's where filters are designed to be *steerable* with respect to the action of $G$ (Cohen and Welling, 2017; Weiler et al., 2018). Concretely, steerable feature fields associate a $D$-dimensional feature vector to each point in a base space $f : \mathbb{R}^2 \to \mathbb{R}^D$ which transform according to their *induced representation* $\left[\mathrm{Ind}_G^{(\mathbb{R}^2 \rtimes G)} \rho\right]$,

$$f(x) \to \left(\left[\mathrm{Ind}_G^{(\mathbb{R}^2 \rtimes G)} \rho\right](tg) \cdot f\right)(x) := \rho(g) \cdot f(g^{-1}(x - t)). \tag{4}$$

Clearly, a RGB image—a scalar field—transforms according to the *trivial representation* $\rho(g) = 1$, $\forall g \in G$, but intermediate layers may transform according to other representation types such as regular. As proven in Cohen et al. (2019), any equivariant linear map between steerable feature spaces transforming under $\rho_i$ and $\rho_{i+1}$ must be a group convolution with $G$-steerable kernels satisfying the following constraint: $\pi_i(gx) = \rho_{i+1}(g)\pi_i(x)\rho_i(g^{-1}) \, \forall g \in G, x \in \mathbb{R}^2$. An equivariant basis is then composed of convolutions with a basis of equivariant kernels that we compute next.

One of the key ingredients needed to apply Theorem 1 is the availability of an equivariant basis with an identity element. One could in principle always take an existing equivariant basis, such as the one provided by Weiler and Cesa (2019), and include an identity element by replacing the first basis element resulting in another equivariant basis with probability 1. In what follows, we show the generality of Theorem 1 by constructing a different equivariant basis from first principles via the canonical basis and then symmetrizing using the action of $G \leq \mathrm{O}(2)$. As we show in our experiments, we can find winning tickets for both basis with negligible difference in performance.

**Classification of Equivariant Maps for E(2).** We now seek to precisely characterize which kernels satisfy the equivariance constraint. Let $\mathcal{R}$ be the equivalence relation on $\mathbb{R}^2$,

$$\mathcal{R} := \forall (x, y) \in \mathbb{R}^2 \times \mathbb{R}^2, \ x \sim y \iff \exists g \in G \quad \text{such that } y = g \cdot x. \tag{5}$$

The equivalence class of $x \in \mathbb{R}^2$ denoted $\mathcal{O}(x)$, is the orbit of $x$ under the action of $G$ on $\mathbb{R}^2$. Designate $\mathcal{A}_\mathcal{R} = \mathbb{R}^2/\mathcal{R} \subset \mathbb{R}^2$ a set of representatives. Due to the equivariance constraint on the kernels $\pi(\cdot)$, once the value of $\pi(x)$ is chosen, it automatically fixes $\pi(g \cdot x)$ for $g \cdot x \in \mathcal{O}(x)$. Note that because $|\mathcal{O}(x)| = |G|$, all possible initial matrices $\mathbb{R}^{|G| \times |G|}$ can be chosen at a point $x \neq 0$.[2]

**Remark**. In practice, $G$-steerable equivariant networks do not operate on signals in $\mathbb{R}^2$ but on a fixed size pixelized grid $\{1, 2, \dots, d\}^2$ denoted as $[d]^2 \subset \mathbb{Z}^2$. Henceforth, we consider all our target networks as well as the overparameterized $G$-steerable network to be defined on input signals sampled on $[d]^2$ and in appendix §E.3 we highlight two practical challenges that result from such a discretization, but crucially these do not disrupt our subsequent theory nor pruning techniques.

---

[2]Care must be taken at the origin, since $\forall g \in G, g \cdot 0 = 0$, and the set of permissible matrices depends on $G$ as well as our choice of representations. We provide a thorough treatment of this case in §E.2.

**Computing $\mathcal{B}$.**[3] To explicitly build a basis of the $G$-equivariant layers, it is illustrative to first consider the case for a single input-output pair of representations for a layer—i.e. $n_i = n_{i+1} = 1$.

We must first construct a basis of the equivariant kernels in our domain. Let $\mathcal{B} = \{\mathcal{B}_x, \ x \in \mathcal{A}_\mathcal{R}\}$ be a basis of equivariant kernels over the domain where each basis is a tensor of shape $\mathcal{B}_x \subset \mathbb{R}^{d \times d \times |G| \times |G|}$. A single basis element $b \in \mathcal{B}_x$ can be constructed by considering the canonical basis[4] $\kappa_0 \subset \mathbb{R}^{|G| \times |G|}$ at each location $x \in \mathcal{A}_\mathcal{R}$ and evaluating it under the action of the group. One can freely choose both a starting point $x \in \mathcal{A}_\mathcal{R}$, and an element of the canonical basis $\kappa_0^{p,q}$. Let $\mathbf{K}_{0,x}^{p,q} \in \mathbb{R}^{d \times d \times |G| \times |G|}, \forall (p,q) \in [|G|] \times [|G|]$ be the tensor of $\kappa_0^{p,q}$ stacked across the grid—i.e. it is 0 everywhere except at the index $(x, p, q)$ where it is 1. Then to get the

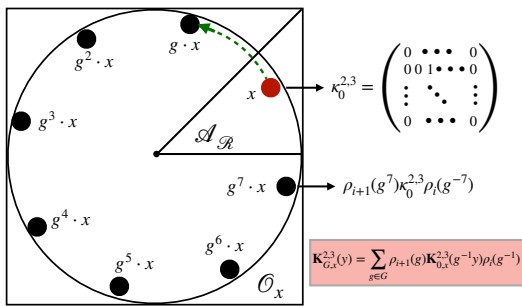

Figure 2: Constructing $\mathbf{K}_{G,x}^{2,3} \in \mathcal{B}_x$ for $C_8$.

equivariant basis we symmetrize by acting on $\mathbf{K}_{0,x}^{p,q}$ while enforcing the equivariance constraint.

$$\forall y \in [d]^2 \quad b(y) := \mathbf{K}_{G,x}^{p,q}(y) = \sum_{g \in G} \rho_{i+1}(g) \mathbf{K}_{0,x}^{p,q}(g^{-1}y) \rho_i(g^{-1}). \tag{6}$$

Repeating this procedure for all elements $\kappa_0^{p,q} \in \mathbb{R}^{|G| \times |G|}$ in the canonical basis completes the construction of our basis $\mathcal{B}_x = \{\mathbf{K}_{G,x}^{p,q}, \ p, q \in [|G|]\}$. We finally obtain a basis of the equivariant kernel as $\mathcal{B} = \bigcup_{x \in \mathcal{A}_\mathcal{R}} \mathcal{B}_x$. A $G$-steerable expanded kernel $\mathbf{K}$ is then simply a linear combination of learned weights $\theta = [\theta_1, \ldots, \theta_{|\mathcal{B}|}]$—one for each basis element—$\mathbf{K}(x) = \sum_{k=1}^{|\mathcal{B}|} \theta_k b_k(x)$. In contrast, a standard convolution kernel has shape $\mathbb{R}^{d \times d \times c_i \times c_{i+1}}$ which means that the equivalent input/output channels for $G$-steerable convolutions are $c_i = |G| \times n_i$ and $c_{i+1} = |G| \times n_{i+1}$ respectively. Fig. 2 illustrates the above process for a basis element for the $C_8$ group. Equipped with this basis, which has an identity element at the origin ( §E.2). We can now apply Thm. 1 to get:

**Corollary 1.** *Let $h \in \mathcal{H}$ be a random $G$-steerable CNN with regular representation of depth $2l$, i.e., $h(\mathbf{x}) = \mathbf{K}_{2l-1}^h * \sigma\left(\ldots \sigma(\mathbf{K}_0^h * \mathbf{x})\right)$ where $\mathbf{K}_{2i}^h \in \mathbb{R}^{d^2 \times |G|^2 \times n_i \times \tilde{n}_i}$, $\mathbf{K}_{2i+1}^h \in \mathbb{R}^{d^2 \times |G|^2 \times \tilde{n}_i \times n_{i+1}}$ are equivariant kernels whose decomposition in $\mathcal{B}$ have coefficients drawn from $\mathcal{U}([-1,1])$. If $\tilde{n}_i = C_3 n_i \log\left(\frac{n_i n_{i+1} d^2 |G|^3 l}{\min(\epsilon, \delta)}\right)$, then with probability at least $1 - \delta$ we have that for all $f \in \mathcal{F}$ (whose kernels $\mathbf{K}_i^f$ have parameters less than 1, and with $\|f_i\| \leq 1$) there exists a collection of pruning masks $\mathbf{S}_{2l-1}, \ldots, \mathbf{S}_0$ such that, by defining $\tilde{\mathbf{K}}_i^h$ the kernel associated with $\mathbf{S}_i \odot \mathbf{W}_i^h$,*

$$\max_{\mathbf{x} \in \mathbb{R}^{d^2 \times n_0}, \|\mathbf{x}\| \leq 1} \|\tilde{\mathbf{K}}_{2l-1}^h * \sigma\left(\ldots \sigma(\tilde{\mathbf{K}}_0^h * \mathbf{x})\right) - f(\mathbf{x})\| \leq \epsilon \tag{7}$$

In Appendix §E, we compute $\max(|\mathcal{B}_{i \to i+1}|, \|\mathcal{B}_{i \to i+1}\|)$ that leads to the corollary above.

### 4.3 SLT FOR PERMUTATION EQUIVARIANT NETS

The symmetric group $\mathcal{S}_n$ consists of all permutations that can be enacted on a set of cardinality $n$. The action of $\mathcal{S}_n$ on a tensor $\mathbf{X} \in \mathbb{R}^{n^k \times m}$ is defined by permuting all but last index: $(g \cdot \mathbf{X})_{i_1, \ldots, i_k, j} = (\mathbf{X}_{g^{-1}(i_1), \ldots, g^{-1}(i_k)}, j), \forall g \in \mathcal{S}_n$. Any general linear permutation equivariant map $\mathbf{W}_i : \mathbb{R}^{n^{k_i}} \to \mathbb{R}^{n^{k_{i+1}}}$, must satisfy the following fixed point equation: $\mathbf{P}^{\otimes(k_i + k_{i+1})} \text{Vec}(\mathbf{W}_i) = \mathbf{W}_i$, where $\mathbf{P}^{\otimes(k_i + k_{i+1})}$ is the $(k_i + k_{i+1})$ Kroenecker power of a permutation matrix $\mathbf{P}$ (Maron et al., 2019). General permutation equivariant networks are the concatenation of linear equivariant layers followed by pointwise non-linearities, which aligns with the setting needed to apply Theorem 1.

---

[3]$\mathcal{B}$ is the basis of equivariant kernels. $\mathcal{B}_{i \to i+1}$ is obtained by taking the 2D convolution with these elements.

[4]Note that this canonical basis is of the same form (but different shape) as $\kappa_{n_i \to n_{i+1}}$ used for $\mathbb{F}_i^{n_i} \to \mathbb{F}_{i+1}^{n_{i+1}}$.

**Classification of all Linear Permutation Equivariant Maps**. In Maron et al. (2019), the authors solve the above fixed point equation by first defining the equivalence relation $\mathcal{Q}$ on $[n]^{k_i + k_{i+1}}$ as:

$$\mathcal{Q} := \forall a, b \in [n]^{k_i + k_{i+1}}, \, a \sim b \Leftrightarrow (\forall i, j \in [k_i + k_{i+1}], a_i = a_j \Leftrightarrow b_i = b_j). \tag{8}$$

Now for all $\mu \in [n]^{k_i + k_{i+1}} / \mathcal{Q}$ define the matrix $B^\mu \in \mathbb{R}^{n^{k_i} \times n^{k_{i+1}}}$ such that each entry $B^\mu_{a,b} = \mathbb{1}_{(a,b) \in \mu}$ [5]. Then a basis for equivariant maps is $\mathcal{B}_{i \to i+1} = \{B^\mu, \mu \in [n]^{k_i + k_{i+1}} / \mathcal{Q}\}$. The cardinality of this basis $|\mathcal{B}_{i \to i+1}| = \tilde{b}(k_i + k_{i+1})$ is known as the $(k_i + k_{i+1})$-th Bell number and can be understood as the number of ways to partition $[n]^{k_i + k_{i+1}}$. When $k_i = k_{i+1}$, the identity element is not in the basis, therefore we replace $B^{(1,\ldots,1)}$ by $\sum_{a \in [n]^k / \mathcal{Q}} B^{(a,a)} = \mathbb{I}$, which is still a basis. We are now in a position to apply Theorem 1 to permutation equivariant networks.

> **Corollary 2.** *Let $h \in \mathcal{H}$ be a random permutation equivariant network of depth $2l$, i.e., $h(\mathbf{x}) = \mathbf{W}^h_{2l-1} \sigma \left( \ldots \sigma(\mathbf{W}^h_0 \mathbf{x}) \right)$ where $\mathbf{W}^h_{2i} \in \mathbb{R}^{n^{k_i} \times n_i \times n^{k_i} \times \tilde{n}_i}$, $\mathbf{W}^h_{2i+1} \in \mathbb{R}^{n^{k_i} \times \tilde{n}_i \times n^{k_{i+1}} \times n_{i+1}}$ are equivariant layers whose decomposition in $\mathcal{B}$ have coefficients drawn from $\mathcal{U}([-1, 1])$. If $\tilde{n}_i = C_2 n_i \log \left( \frac{n_i n_{i+1} \max(\tilde{b}(k_i + k_{i+1}), n^{k_i} + 1)l}{\min(\epsilon, \delta)} \right)$, then with probability at least $1 - \delta$ we have that for all $f \in \mathcal{F}$ (with $\|\|f_i\|\| \leq 1$ and parameters in the basis less than 1) there exists a collection of pruning masks on the decomposition in the equivariant basis of the layers $\mathbf{S}_{2l-1}, \ldots, \mathbf{S}_0$ s.t.,*
>
> $$\max_{\mathbf{x} \in \mathbb{R}^{n^{k_0} \times n_0}, \|\mathbf{x}\| \leq 1} \|(\mathbf{S}_{2l-1} \odot \mathbf{W}^h_{2l-1})\sigma \ldots \sigma((\mathbf{S}_0 \odot \mathbf{W}^h_0)(\mathbf{x})) - f(\mathbf{x})\| \leq \epsilon \tag{9}$$

We discuss in Appendix §F.1 the computation of $\|\|\mathcal{B}_{i \to i+1}\|\|$, and provide the detailed proof.

**Message Passing GNNs**. MPGNNs are networks that act on graphs with $n$-nodes by defining a feature vector for each node which is updated based on "messages" received from its neighbors which are then combined. Given a node $v$ in a graph and its hidden representation $x^v_i$, the message passing update for a layer $i$ is governed by the following equation: $x^v_i = f^{\text{up}}_i(x^v_{i-1}, \sum_{u \in \mathcal{N}(v)} f^{\text{agg}}_i(x^v_{i-1}, x^u_{i-1}))$. In its most general form the aggregation function $f^{\text{agg}}_i$ and update function $f^{\text{up}}_i$ are taken to be MLPs. In this case it is easy to see that Theorem 1 can be applied separately to both $f^{\text{agg}}_i$, $f^{\text{up}}_i$ independently as MLPs are captured under $G = \{e\}$. Permutation in/equivariance is trivially maintained in the pruned network as the aggregate function operates on a local neighborhood of $v$ and pruning does not impact this as pruning does not impose any ordering over the nodes or the adjacency matrix in the graph.

## 5 EXPERIMENTS

We substantiate our equivariant framework to finding winning SLTs by approximating target $G$-steerable networks, MPGNNs, and $k$-order GNNs on standard image classification, node and graph classification tasks respectively. For steerable networks we consider networks for $G \in \{C_4, C_8, D_4\}$ which are finite subgroups of O(2). To show the generality of our framework, we experiment with two different equivariant basis for E(2); the first one uses spherical harmonics and is taken from Weiler and Cesa (2019) (DEFAULT), while the second is the one we introduce in §4.2 (OURS). MPGNNs and $k$-order GNNs naturally operate on $\mathcal{S}_n$ where permutation invariance is with respect to the node labels of a given graph. For E(2)-steerable, we experiment with Rotation and FlipRotation-MNIST datasets which contain data augmentations from $G \leq \text{SO}(2)$ and $G \leq \text{O}(2)$ respectively (Weiler and Cesa, 2019). To evaluate MPGNNs and $k$-order GNNs we consider standard node classification benchmarks in citation networks in Cora and CiteSeer (Sen et al., 2008) and real-world graph classification datasets in Proteins and NCI1 (Yanardag and Vishwanathan, 2015).

We find equivariant strong lottery tickets by utilizing our overparametrization strategy described in §3 by solving SUBSET-SUM problems using Gurobi (Gurobi Optimization, 2018). The definition of the SUBSET-SUM problems as mixed-integer optimization problems can be found in eq. 28 of §G. In Table 3 we report our main results for an overparametrization constant $C = 5$ (see Thm. 1) towards approximating a single target network using 5 random seeds to construct our overparametrized network. Specifically, we report the ratio of the number of parameters in the overparametrized and

---

[5] $\mathbb{1}_{(a,b) \in \mu} = 1$ if $(a, b) \in \mu$ and 0 otherwise, for $a \in [n]^{k_i}$ and $b \in [n]^{k_{i+1}}$

final pruned network divided by the original target network. We also report test accuracies for both, the maximum absolute weight error over all SUBSET-SUM problems, and the maximum relative output error between pruned and target networks. All model architectures and described in §G.

For all equivariant architectures and datasets considered, we find that we are able to approximate the corresponding trained target networks sufficiently well. Specifically, we achieve sufficiently low maximum relative output error across test samples such that the test accuracy of the resulting pruned network matches essentially that of the target one for all random seeds of the pruning experiments. Finally, we conduct an ablation study on the effect of overparametrization constant factor $C$ to the approximation accuracy with respect to the tolerance $\epsilon$. We perform this study for the E(2) equivariant architectures for different subgroups. In Fig. 3 we plot this as a function of $C \in \{1, 2, 5, 10\}$ for the groups $C_4, C_8, D_4$ using the basis construction from Weiler and Cesa (2019). As observed, increasing our overparametrization factor leads, up to $C = 5$, to a lower maximum relative output error while the pruned accuracy marginally increases.

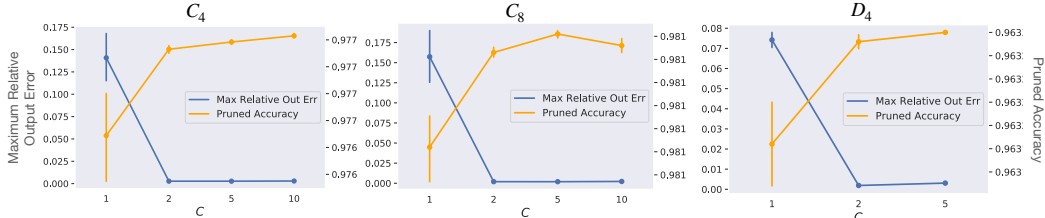

Figure 3: Ablation study of max. relative output error and pruned accuracy w.r.t. to $C$ for $C_4, C_8, D_4$.

| Task | | Ratio $p/p_{\text{target}}$ | | Accuracy (%) | | Errors | |
|---|---|---|---|---|---|---|---|
| Arch. | Dataset | overparam. | pruned | target | pruned[†] | param. | output |
| MPGNN | Cora | $2.0e^4$ | 99.1 | 80.2 | 80.2 | $6.2e^{-3} \pm 3.3e^{-3}$ | $3.7e^{-4} \pm 0.5e^{-4}$ |
| | CiteSeer* | $5.5e^4$ | 103.8 | 63.4 | 63.4 | $1.1e^{-1} \pm 1.7e^{-1}$ | $1.8e^{-3} \pm 1.7e^{-3}$ |
| $k$-order GNN* | Proteins | $3.6e^2$ | 51.8 | 81.08 | 81.08 | $1.2e^{-4} \pm 4.2e^{-5}$ | $8.8e^{-4} \pm 2.9e^{-4}$ |
| | NCI1 | $5.3e^2$ | 85.9 | 74.21 | 74.21 | $8.1e^{-2} \pm 5.2e^{-2}$ | $2.6e^{-2} \pm 9.9e^{-3}$ |
| E2-C4-DEFAULT | | $3.0e^2$ | 41.0 | 97.7 | 97.7 | $1.4e^{-3} \pm 0.0e^{-3}$ | $2.7e^{-3} \pm 0.4e^{-3}$ |
| E2-C8-DEFAULT | Rot- | $3.1e^2$ | 41.2 | 98.1 | 98.1 | $1.2e^{-3} \pm 0.2e^{-3}$ | $1.9e^{-3} \pm 0.2e^{-3}$ |
| E2-C4-OURS* | MNIST | $2.9e^2$ | 122.3 | 96.2 | 96.2 | $6.2e^{-1} \pm 5.8e^{-1}$ | $2.4e^{-2} \pm 0.3e^{-2}$ |
| E2-C8-OURS* | | $3.0e^2$ | 120.9 | 96.8 | 96.8 | $2.1e^{-2} \pm 0.6e^{-2}$ | $3.3e^{-1} \pm 3.7e^{-1}$ |
| E2-D4-DEFAULT | FlipRot- | $3.1e^2$ | 77.0 | 96.3 | 96.3 | $4.1e^{-2} \pm 4.2e^{-2}$ | $1.4e^{-2} \pm 1.1e^{-2}$ |
| E2-D4-OURS | MNIST | $3.1e^2$ | 115.4 | 94.1 | 94.1 | $2.4e^{-1} \pm 0.9e^{-1}$ | $4.2e^{-2} \pm 1.2e^{-2}$ |

Table 3: Pruning random overparameterized $G$-equivariant networks to approximate $G$-equivariant targets. We report a) $p/p_{\text{target}}$ the parameter ratio of the number of parameters $p$ of the overparametrized or the final pruned networks over $p_{\text{target}}$, b) the test accuracy of the target and the pruned networks, c) the maximum absolute weight error over subset sum problems, d) and the relative output errors of the pruned network in contrast to the target over samples in the test set. [†]STDs are below $1e^{-4}$. *Maximum time of MIP solver for SUBSET-SUM problems was thresholded to 600ms.

## 6 DISCUSSION

This paper introduces a unifying framework to prove the strong lottery ticket hypothesis for general equivariant networks. We prove the existence with high probability of winning tickets for randomly (logarithmically) overparameterized networks with double the depth. We also theoretically demonstrate such an overparametrization scheme is optimal as a function of the tolerance. While our presented theory is built using overparametrized networks of depth $2L$ it may be possible to extend Theorem 1 to the setting where overparamatrized networks have depth $L + 1$ as in Burkholz (2022b) by adapting the proof techniques. We leave this extension as future work. Our framework enjoys broad applicability to MLPs, CNNs, E(2)-steerable networks, general permutation equivariant networks, and MPGNNs all of which become insightful corollaries of our main theoretical result. One limitation of our developed theory is the assumption of using a point-wise ReLU as the non-linearity. As a result, a natural direction for future work is to consider extensions of the SUBSET-SUM problem beyond linear functions to more general non-linearities. In addition, our overparametrization strategy employed the "diamond shape" technique; however other schemes might also yield an optimal upper bound. Characterizing these schemes is an exciting direction for future work.

## 7 ETHICS STATEMENT

The main contributions of this work are primarily theoretical in nature as we seek to provide a general framework to study equivariant lottery tickets. Consequently, any potential societal impact would necessarily be speculative in nature and deeply tied to a particular application domain. For example, one could consider the environmental cost savings from creating an overparametrized $G$-equivariant network that does not need any GPU hours to train, but instead CPU resources to solve SUBSET-SUM problems. Beyond these goals any application of our theory to actual practice is likely to inherit the complex broader impacts native to the problem domain and we encourage practitioners to exercise due caution in their efforts.

## 8 REPRODUCIBILITY STATEMENT

We provide a complete proofs for all our theoretical results in the Appendix. In particular, proofs for Lemma 1 can be found in Appendix B.1 and Theorem 1 is a direct application of this result $l$-times and whose proof is located in B.2. The proof for Theorem 2 is located in Appendix B.3. Furthermore, instantiations of framework for E(2)-steerable CNNs, permuation equivariant networks, MLPs, and vanilla CNNs resulting in corollaries 1, 2, 3, and 4 respectively. The proofs for all the corollaries are located in Appendices C (MLP), D (CNN), E.4 (E(2)-CNN), and F.1 (permutation equivariant networks). We provide full details on our experimental setup, including hyperparamters choices, architectures, and the exact SUBSET-SUM problem being solved for pruning in Appendix G. Finally, code to reproduce our experimental results can be found in submission's supplementary material.

## 9 ACKNOWLEDGEMENTS

The authors would like to thank Louis Pascal Xhonneux, Mandana Samiei, Mehrnaz Mofakhami, and Tara Akhound-Sadegh for insightful feedback on early drafts of this work. In addition, the authors thank Riashat Islam, Manuel Del Verme, Mandana Samiei, and Andjela Mladenovic for their generous sharing of computational resources. AJB is supported by the IVADO Ph.D. Fellowship.

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

## A    ADDITIONAL MATERIAL ON THE SUBSET SUM PROBLEM

We recall here some results on subset sum originally from Lueker (1998) and modified by Pensia et al. (2020) to better fit the proof.

**Lemma 2** (SUBSET-SUM lemma). *Let $U \simeq \mathcal{U}([0,1])$ (or $\mathcal{U}([-1,0])$ and $V \simeq \mathcal{U}([-1,1])$ be two independent random variables. Let $P$ be the distribution of $UV$. Let $\delta_0$ be the dirac-delta function. Define a distribution $D = \frac{1}{2}\delta_0 + \frac{1}{2}P$. Let $X_1, \ldots X_n$ be i.i.d. from the distribution $D$ where $n \geq C \log(\frac{2}{\epsilon})$ (for some universal constant C). Then, with probability at least $1 - \epsilon$, we have*

$$\forall z \in [-1,1], \exists S \subset [n] \quad such\ that \quad |z - \sum_{i \in S} X_i| \leq \epsilon \tag{10}$$

This Lemma, is in fact a consequence of the corollary 3.3 from Lueker (1998) which states that as soon as a distribution contains a uniform distribution, one can achieve any target with exponentially small precision by SUBSET-SUM.

**Extension to more general distributions**.This allows us to extend the result Theorem 1 to a more general setting, where the distribution of the random coefficients is not $\mathcal{U}([-1,1])$ but contains a uniform distribution.

Let's say that a distribution $Z$ contains a uniform distribution $\mathcal{U}([a,b])$ if there exist a distribution $Z_1$ and a constant $\zeta \in [0,1[$ such that:

$$Z := \zeta Z_1 + (1 - \zeta)\mathcal{U}([a,b])$$

We want to extend the results of theorem 1 to distributions containing $\mathcal{U}([-a,a])$ for some $a > 0$

We follow therefore the same path as in Pensia et al. (2020) to prove Lemma 2 but with more general distributions. Pensia et al. (2020) already made a remark for this next extension that we state and prove here.

**Lemma 3.** *Let $a > 0$. Let $X$ and $Y$ be two independent random variables such that $X$ contains $\mathcal{U}([0,a])$ (or $\mathcal{U}([-a,0])$) and $Y$ contains $\mathcal{U}([-a,a])$. Then the PDF of the random variable $XY$ is such that:*

$$\exists A_a > 0, \quad f_{XY}(z) \geq A_a \log\left(\frac{a^2}{|z|}\right) \quad for |z| < a^2$$

*Proof.* By the change of variable $\frac{X}{a}$ and $\frac{Y}{a}$, one can apply lemma 4 from Pensia et al. (2020) to get that if $\tilde{X} \sim \mathcal{U}([0,a])$ (or $\mathcal{U}([-a,a])$) and $\tilde{Y} \sim \mathcal{U}([-a,a]]$ the PDF of $\tilde{X}\tilde{Y}$ is:

$$\frac{1}{2a^2} log\left(\frac{a^2}{|z|}\right) \text{ if } |z| \leq a^2 \text{ and 0 otherwise}$$

Now we know by hypothesis that $\exists \alpha_X, \alpha_Y > 0$ such that $f_X \geq \alpha_X f_{\mathcal{U}([0,a])}$ and $f_Y \geq \alpha_Y f_{\mathcal{U}([-a,a])}$.

Therefore, $f_{XY} \geq \alpha_X \alpha_Y f_{\tilde{X}\tilde{Y}}$ and finally, $f_{XY} \geq \frac{\alpha_X \alpha_Y}{2a^2} log\left(\left(\frac{a^2}{|z|}\right)\right)$ if $|z| \leq a^2$    □

**Lemma 4.** *Let $X$ and $Y$ be two independent random variables such that $X$ contains $\mathcal{U}([0,a])$ (or $\mathcal{U}([-a,0])$) and $Y$ contains $\mathcal{U}([-a,a])$. Let $P$ be the distribution of $XY$. Then there exists a distribution $Q$ and a scalar $B_a > 0$ such that:*

$$P = B_a \mathcal{U}([-\frac{a^2}{2}, \frac{a^2}{2}]) + (1 - B_a)Q$$

*Proof.* This is a direct consequence of the lower bound on the PDF of $XY$ that was shown in the previous Lemma.    □

Using Lemma 4 and Corollary 3.3 from Lueker (1998) leads immediately to the following result:

**Lemma 5.** *Let $a > 0$, $X$ be a random variable containing $\mathcal{U}([0, a])$ (or $\mathcal{U}([-a, 0])$) and $Y$ containing $\mathcal{U}([-a, a])$. Let $X_1, \ldots, X_n$ be $n$ iid random variables following the distribution $\frac{1}{2}\delta_0 + \frac{1}{2}P$ where $P$ is the distribution of $XY$. Then, if $n \geq C_a \log\left(\frac{2}{\epsilon}\right)$ (for some constant depending on $a$), with probability at least $1 - \epsilon$, we have:*

$$\forall z \in [-1, 1], \exists S \subset \{1, \ldots, n\} \quad |z - \sum_{i \in S} X_i| \leq \epsilon$$

*Proof.* This follows immediately from Corollary 3.3 in Lueker (1998) (by applying Markov's inequality) and Lemma 4. $\square$

**Discussion**. This allows us to generalize Theorem 1 to settings where the random overparametrized network has weights taken from a distribution which contains $\mathcal{U}([-a, a])$. This includes almost all the usual settings, namely Gaussian, uniform, ... Indeed, the only thing to change is to no longer use Lemma 2 but Lemma 5 at the same place in the proof and by assuming that the distribution of the parameters of the overparametrized network contains $\mathcal{U}([-a, a])$ for some $a > 0$.

# B    Proof of the General SLT on equivariant networks using pointwise ReLU

## B.1    Approximation of an equivariant target layer

We now prove Lemma 1 that is used to approximate a single layer in a $G$-equivariant target model.

**Lemma 1.** *Let $h_i \in \mathcal{H}_i$ be a random overparametrized $G$-equivariant network as defined above, with coefficients $\lambda_{p \to q, k}^{(i)}$ and $\mu_{p \to q, k}^{(i)}$ drawn from $\mathcal{U}([-1, 1])$. Further suppose that each $\tilde{n}_i = C_1 n_i \log\left(\frac{n_i n_{i+1} \max(|\mathcal{B}_{i \to i+1}|, \|\|\mathcal{B}_{i \to i+1}\|\|)}{\min(\epsilon, \delta)}\right)$ where $C_1$ is a constant. Then, with probability $1 - \delta$, for every target $G$-equivariant layer $f_i \in \mathcal{F}_i$, one can find two pruning masks $\mathbf{S}_{2i}, \mathbf{S}_{2i+1}$ on the coefficients $\lambda_{p \to q, k}^{(i)}$ and $\mu_{p \to q, k}^{(i)}$ respectively such that:*

$$\max_{\mathbf{x} \in \mathbb{R}^{D_i \times n_i}, \|\mathbf{x}\| \leq 1} \|(\mathbf{S}_{2i+1} \odot \mathbf{W}_{2i+1}^h)\sigma((\mathbf{S}_{2i} \odot \mathbf{W}_{2i}^h)\mathbf{x}) - f_i(\mathbf{x})\| \leq \epsilon. \tag{1}$$

*Proof.* Let us first recall that the main hypothesis needed for this lemma is to have an identity element in the basis $\mathbb{I} \in \mathcal{B}_{i \to i}$. We note that this is a very mild assumption since the identity is trivially equivariant between $\mathbb{F}_i$ and $\mathbb{F}_i$ and one can always choose to incorporate it in the basis. Consequently, we will designate the first element in our equivariant basis to be the identity $b_{i \to i, 1} = \mathbb{I}$.

**Remark.** We choose $C_1 = 3C$ to ensure that ($C$ is the universal constant introduced in lemma 2):

$$C_1 \log\left(\frac{n_i n_{i+1} \max(|\mathcal{B}_{i \to i+1}|, \|\|\mathcal{B}_{i \to i+1}\|\|)}{\min(\epsilon, \delta)}\right) \geq C \log\left(\frac{4 n_i n_{i+1} \max(|\mathcal{B}_{i \to i+1}|, \|\|\mathcal{B}_{i \to i+1}\|\|)}{\min(\epsilon, \delta)}\right),$$

which is true for the entire domain of variables we are interested in ($n_i, n_{i+1} \geq 1, \delta, \epsilon \leq \frac{1}{2}$ and $|\mathcal{B}_{i \to i+1}| \geq 1$). It is easy to see that $3 \log(x) \geq \log(4x)$ on $[2, +\infty[$ as $x^3 \geq 4x$ in this domain.

To begin, we first introduce a function, $\chi$, to identify blocks in our feature space $\mathbb{F}_i^{n_i}$. In particular, we leverage the "diamond shape" structure (see Fig 1) and define $\chi : [\tilde{n}_i] \to [n_i]$, such that it divides the intermediate layer of our overparametrized approximation into groups of $C_1 \log\left(\frac{n_i n_{i+1} \max(|\mathcal{B}_{i \to i+1}|, \|\|\mathcal{B}_{i \to i+1}\|\|)}{\min(\epsilon, \delta)}\right)$ blocks which are linked with the same block in the first (i-th) layer. In other words, $\chi$ associates a block in $\mathbb{F}_i^{\tilde{n}_i}$ in a surjective manner to a block in $\mathbb{F}_i^{n_i}$. In a last piece of notation we will use $x_\omega$ to mean the $\omega$-th block of the feature space for $x$. For example, if $x \in \mathbb{R}^{n_i \times D_i}$ which is contained in the feature space $\mathbb{F}_i^{n_i}$ of the $i$-th layer then $\omega \in [n_i]$ and $x_\omega \in \mathbb{F}_i$

denotes the $\omega$-th vector of dimension $D_i$ in $x$. Finally, because $\omega$ is a dummy index, quite often we will replace it with appropriate layer index—e.g. $p, q, r$. With this in hand we can write the function $\chi(q)$ as follows:

$$\chi(q) = \left\lfloor \frac{q-1}{C_1 \log\left(\frac{n_i n_{i+1} \max(|\mathcal{B}_{i \to i+1}|, \|\mathcal{B}_{i \to i+1}\|)}{\min(\epsilon, \delta)}\right)} \right\rfloor + 1$$

Before pruning, one has

$$\mathbf{W}_{2i}^h = \sum_{p=1}^{n_i} \sum_{q=1}^{\tilde{n}_i} \sum_{k=1}^{|\mathcal{B}_{i \to i}|} (\kappa_{n_i \to \tilde{n}_i}^{p,q} \otimes \lambda_{p \to q, k}^{(i)} b_{i \to i, k}).$$

We begin pruning by annihilating all first layer coefficients not associated with the identity basis element ($k \neq 1$) $\lambda_{p \to q, k}^{(i)}$ and for $q \notin \chi^{-1}(p)$. This yields the following decomposition post-pruning,

$$\mathbf{W}_{2i}^h = \sum_{p=1}^{n_i} \sum_{q \in \chi^{-1}(p)} \left( \kappa_{n_i \to \tilde{n}_i}^{p,q} \otimes \lambda_{p \to q, 1}^{(i)} \mathbb{I} \right).$$

Note that we can write $p = \chi(q)$ leading to the following:

$$\left( \mathbf{W}_{2i}^h x \right)_q = \lambda_{\chi(q) \to q, 1}^{(i)} x_{\chi(q)}.$$

After the $\sigma$, —i.e. the pointwise-ReLU, one then has:

$$\sigma(\mathbf{W}_{2i}^h x)_q = \sigma(\lambda_{\chi(q) \to q, 1}^{(i)} x_{\chi(q)}) = \lambda_{\chi(q) \to q, 1}^{(i)+} x_{\chi(q)}^+ + \lambda_{\chi(q) \to q, 1}^{(i)-} x_{\chi(q)}^-$$

where we used the fact that the ReLU is pointwise and the identity on scalars $\sigma(wx) = w^+ x^+ + w^- x^-$. Expanding the second layer in its equivariant basis and using the above equation we get:

$$\left[ \mathbf{W}_{2i+1}^h \sigma(\mathbf{W}_{2i}^h x) \right]_r = \sum_{q=1}^{\tilde{n}_i} \left( \sum_{k=1}^{|\mathcal{B}_{i \to i+1}|} \mu_{q \to r, k}^{(i)} b_{i \to i+1, k} \right) \sigma(\mathbf{W}_{2i}^h x)_q \tag{11}$$

$$= \sum_{p=1}^{n_i} \sum_{q \in \chi^{-1}(p)} \left( \sum_{k=1}^{|\mathcal{B}_{i \to i+1}|} \mu_{q \to r, k}^{(i)} b_{i \to i+1, k} \right) (\lambda_{p \to q, 1}^{(i)+} x_p^+ + \lambda_{p \to q, 1}^{(i)-} x_p^-) \tag{12}$$

$$= \sum_{p=1}^{n_i} \sum_{k=1}^{|\mathcal{B}_{i \to i+1}|} \sum_{q \in \chi^{-1}(p)} \left( \mu_{q \to r, k}^{(i)} \lambda_{p \to q, 1}^{(i)+} \right) b_{i \to i+1, k} x_p^+ + \tag{13}$$

$$\sum_{p=1}^{n_i} \sum_{k=1}^{|\mathcal{B}_{i \to i+1}|} \sum_{q \in \chi^{-1}(p)} \left( \mu_{q \to r, k}^{(i)} \lambda_{p \to q, 1}^{(i)-} \right) b_{i \to i+1, k} x_p^-. \tag{14}$$

Our goal is to approximate the target model whose $r$-th block can be written as:

$$f_i(x)_r = \sum_{p=1}^{n_i} \sum_{k=1}^{|\mathcal{B}_{i \to i+1}|} \underbrace{\alpha_{p \to r, k}^{(i)} b_{i \to i+1, k} x_p^+}_{\text{term 1}} + \sum_{p=1}^{n_i} \sum_{k=1}^{|\mathcal{B}_{i \to i+1}|} \underbrace{\alpha_{p \to r, k}^{(i)} b_{i \to i+1, k} x_p^-}_{\text{term 2}}.$$

To do so, we only have to approximate $\alpha^{(i)}_{p \to r,k}$ in term 1, for all $p, r, k$, using a subset sum of $\sum_{q \in \chi^{-1}(p)} \left( \mu^{(i)}_{q \to r,k} \lambda^{(i)+}_{p \to q,1} \right)$ and $\alpha^{(i)}_{p \to r,k}$ in term 2, by a subset sum of $\sum_{q \in \chi^{-1}(p)} \left( \mu^{(i)}_{q \to r,k} \lambda^{(i)-}_{p \to q,1} \right)$. This can be achieved by judiciously choosing pruning masks that selectively include $\mu^{(i)}_{q \to r,k}$ which is a by-product of solving independent SUBSET-SUM problems.

The key insight powering our analysis is to notice that the variables $\mu^{(i)}_{p \to r,k}$ that appear in each approximation problems are different if $(p, r, k) \neq (p', r', k')$. Moreover, the two different problems for fixed indices $(p, r, k)$ can be seen using different variables since following whether it is positive or negative, $\lambda^{(i)}_{p \to q,1}$ will necessarily be 0 in the first or the second term equation. Therefore, either in the first or the second equation, $\mu^{(i)}_{q \to r,k}$ can be seen as being not a variable of the SUBSET-SUM problem. We are then at liberty to decide whether to prune the variable or not in the equation where it appears, because the pruning of the variable will not affect the result of the other SUBSET-SUM problem. Following this approach, we can then find a mask on the variables implied in subsequent problems, solve the problems independently and finally take the concatenation of all the masks in the second layer which will simultaneously solve all the problems.

We now quantify this approach by showing that with high probability, the $2 n_i n_{i+1} |\mathcal{B}_{i \to i+1}|$ subset sum problems (with independent variables) written below can all be solved by applying a pruning mask on the second layer. The pruned mask applied on the second layer is denoted $\mathbf{S}^{2i+1}_{q \to r,k} \in \{0, 1\}^{\tilde{n}_i \times n_{i+1} \times |\mathcal{B}_{i \to i+1}|}$. The subset sum problems are written below:

$$|err^{(i)}_{p \to r,k,+}| := \left| \sum_{q \in \chi^{-1}(p)} (\mathbf{S}^{2i+1}_{q \to r,k} \circ \mu^{(i)}_{q \to r,k}) \lambda^{(i)+}_{p \to q,k} - \alpha^{(i)}_{p \to r,k} \right| \quad \forall (p, r, k) \in [n_i] \times [n_{i+1}] \times [|\mathcal{B}_{i \to i+1}|]$$

$$\leq \frac{\epsilon}{2 n_i n_{i+1} \max(|\mathcal{B}_{i \to i+1}|, \|\mathcal{B}_{i \to i+1}\|)}$$

and

$$|err^{(i)}_{p \to r,k,-}| := \left| \sum_{q \in \chi^{-1}(p)} (\mathbf{S}^{2i+1}_{q \to r,k} \circ \mu^{(i)}_{q \to r,k}) \lambda^{(i)-}_{p \to q,k} - \alpha^{(i)}_{p \to r,k} \right| \quad \forall (p, r, k) \in [n_i] \times [n_{i+1}] \times [|\mathcal{B}_{i \to i+1}|]$$

$$\leq \frac{\epsilon}{2 n_i n_{i+1} \max(|\mathcal{B}_{i \to i+1}|, \|\mathcal{B}_{i \to i+1}\|)}$$

We will now use the SUBSET-SUM Lemma 2 which explains the overparametrization that one needs to solve the SUBSET-SUM problems. Since $\mu^{(i)}_{q \to r,k}$ and $\lambda^{(i)}_{p \to q,k}$ are i.i.d following $\mathcal{U}([-1,1])$, $\lambda^{(i),+}_{p \to q,1}$ follows $\frac{1}{2}\delta_0 + \frac{1}{2}U$ with the notations of lemma 2. We deduce that the $\mu^{(i)}_{q \to r,k} \lambda^{(i),+}_{p \to q,1}$ are i.i.d. following the distribution $D = \frac{1}{2}\delta_0 + \frac{1}{2}P$. This is the same for $\mu^{(i)}_{q \to r,k} \lambda^{(i),-}_{p \to q,1}$ which are i.i.d. following the distribution $D = \frac{1}{2}\delta_0 + \frac{1}{2}P$. Here one should note that $\forall p \in [n_i], |\chi^{-1}(p)| = C_1 \log(\frac{n_i n_{i+1} \max(|\mathcal{B}_{i \to i+1}|, \|\mathcal{B}_{i \to i+1}\|)}{\min(\epsilon, \delta)})$. Therefore, by using[6] lemma 2, $\forall (p, r, k) \in [n_i] \times [n_{i+1}] \times [|\mathcal{B}_{i \to i+1}|]$, the two subset sum problems can be achieved by pruning the coefficients $\mu^{(i)}_{q \to r,k}$ with probability at least $1 - \frac{\delta}{2 n_i n_{i+1} \max(|\mathcal{B}_{i \to i+1}|, \|\mathcal{B}_{i \to i+1}\|)}$. Call this the event $E^{(i)}_{p \to r,k}$. By taking the intersection of the events, we get that $E^{(i)} = \bigcap_{(p,r) \in [n_i] \times [n_{i+1}], k \in [|\mathcal{B}_{i \to i+1}|]} E^{(i)}_{p \to r,k}$ holds with probability at least,

$$p(E^{(i)}) = 1 - n_i n_{i+1} |\mathcal{B}_{i \to i+1}| \frac{\delta}{2 n_i n_{i+1} \max(|\mathcal{B}_{i \to i+1}|, \|\mathcal{B}_{i \to i+1}\|)}$$

$$\geq 1 - \delta.$$

---

[6] At this point one may prove the same lemma but with more general distributions on the coefficients $\lambda^{(i)}_{p \to q,k}$ and $\mu^{(i)}_{q \to r,k}$ by assuming that they only contain $\mathcal{U}([-a, a])$ for some $a > 0$ and by using Lemma 5 instead of Lemma 2

In other words, with probability at least $1 - \delta$, all the SUBSET-SUM problems are solved. Finally, it remains to check that the approximation holds with this pruning mask. Let $\Omega$ be defined as:

$$\Omega = \max_{\|x\| \leq 1} \|(\mathbf{S}_{2i+1} \odot \mathbf{W}_{2i+1}^h)\sigma((\mathbf{S}_{2i} \odot \mathbf{W}_{2i}^h)\mathbf{x}) - f_i(\mathbf{x})\|.$$

By applying the masks we get:

$$\Omega = \max_{r \in [n_{i+1}]} \max_{\|x\| \leq 1} \left\|(\mathbf{S}_{2i+1} \odot \mathbf{W}_{2i+1}^h)\sigma((\mathbf{S}_{2i}^h \odot \mathbf{W}_{2i}^h)\mathbf{x})_r - f_i(\mathbf{x})_r\right\|$$

$$= \max_{r \in [n_{i+1}]} \max_{\|x\| \leq 1} \left\| \sum_{p=1}^{n_i} \sum_{k=1}^{|\mathcal{B}_{i\to i+1}|} \left((\alpha_{p\to r,k}^{(i)} + err_{p\to r,k,+}^{(i)})b_{i\to i+1,k}(x_p^+) + (\alpha_{p\to r,k}^{(i)} + err_{p\to r,k,-}^{(i)})b_{i\to i+1,k}(x_p^-)\right) \right.$$

$$\left. - \sum_{p=1}^{n_i} \sum_{k=1}^{|\mathcal{B}_{i\to i+1}|} \left(\alpha_{p\to r,k}^{(i)}b_{i\to i+1,k}(x_p^+) + \alpha_{p\to r,k}^{(i)}b_{i\to i+1,k}(x_p^-)\right) \right\|$$

$$\leq \max_{r \in [n_{i+1}]} \sum_{p=1}^{n_i} \max_{\|x\| \leq 1} \left\| \sum_{k=1}^{|\mathcal{B}_{i\to i+1}|} err_{p\to r,k,+}^{(i)}b_{i\to i+1,k}(x_p^+) + \sum_{k=1}^{|\mathcal{B}_{i\to i+1}|} err_{p\to r,k,-}^{(i)}b_{i\to i+1,k}(x_p^-) \right\|$$

$$\leq \max_{r \in [n_{i+1}]} \sum_{p=1}^{n_i} \left( \max_{\|x\| \leq 1} \left\| \sum_{k=1}^{|\mathcal{B}_{i\to i+1}|} err_{p\to r,k,+}^{(i)}b_{i\to i+1,k}(x_p^+) \right\| + \max_{\|x\| \leq 1} \left\| \sum_{k=1}^{|\mathcal{B}_{i\to i+1}|} err_{p\to r,k,-}^{(i)}b_{i\to i+1,k}(x_p^-) \right\| \right)$$

$$\leq \max_{r \in [n_{i+1}]} \sum_{p=1}^{n_i} \left( \left\|\left\| \sum_{k=1}^{|\mathcal{B}_{i\to i+1}|} err_{p\to r,k,+}^{(i)}b_{i\to i+1,k}(x_p^+) \right\|\right\| + \left\|\left\| \sum_{k=1}^{|\mathcal{B}_{i\to i+1}|} err_{p\to r,k,-}^{(i)}b_{i\to i+1,k}(x_p^-) \right\|\right\| \right)$$

$$\leq \sum_{p=1}^{n_i} \frac{2\epsilon}{2n_i n_{i+1} \max(|\mathcal{B}_{i\to i+1}|, |\!|\!|\mathcal{B}_{i\to i+1}|\!|\!|)} \times |\!|\!|\mathcal{B}_{i\to i+1}|\!|\!|$$

$$\leq \epsilon$$

$\square$

**Note**. In the statement of the Lemma we used a specific choice of norm ($l_p$) but our proof strategy will work with every norm as soon as $\sigma$, the ReLU non-linearity, is 1-Lipschitz (which may not be the case for some esoteric norms). As a result, there is no need to restrict oneself to the $l_p$-norm, though for ease of exposition and not to confuse the reader we made this choice above. Moreover thanks to the flexibility of the SUBSET-SUM theorem, the proof can also be extended to a milder hypothesis which is on the distribution of coefficients. Specifically, it is sufficient to have that the distribution contains a uniform distribution centered at $0$ (see Lemma 5). The immediate consequence of this is that it is possible to accommodate other weight initialization schemes that are commonly used in practice, but again for ease of readibility we chose to use Uniform distribution.

## B.2 APPROXIMATION OF AN EQUIVARIANT TARGET NETWORK

We now prove in this appendix Theorem 1 which approximates a full target model. We first recall the two main assumptions (very mild) that are needed for the Theorem statement:

- $\mathbb{I} \in \mathcal{B}_{i\to i}$
- $\sigma$ the pointwise ReLU is used as an equivariant nonlinearity and is 1-Lipschitz.

**Theorem 1.** *Let $h \in \mathcal{H}$ be a random overparametrized G-equivariant network with coefficients $\lambda_{p\to q,k}^{(i)}$ and $\mu_{p\to q,k}^{(i)}$, for $i \in [l]$ and indices $p, q, k$ as defined in Table 1, all drawn from $\mathcal{U}([-1,1])$. Suppose that $\tilde{n}_i = C_2 n_i \log(\frac{n_i n_{i+1} \max(|\mathcal{B}_{i\to i+1}|, |\!|\!|\mathcal{B}_{i\to i+1}|\!|\!|)l}{\min(\epsilon,\delta)})$, where $C_2$ is a constant. Then with probability $1 - \delta$, for every $f \in \mathcal{F}$, one can find a collection of pruning*

*masks $\mathbf{S}_{2l-1}, \ldots \mathbf{S}_0$ on the coefficients $\lambda_{p \to q, k}^{(i)}$ and $\mu_{p \to q, k}^{(i)}$ for every layer $i \in [l]$ such that:*

$$\max_{\mathbf{x} \in \mathbb{R}^{D_0 \times n_0}, \, \|\mathbf{x}\| \leq 1} \|(\mathbf{S}_{2l-1} \odot \mathbf{W}_{2l-1}^h)\sigma \left( \ldots \sigma((\mathbf{S}_0 \odot \mathbf{W}_0^h)\mathbf{x}) \right) - f(\mathbf{x})\| \leq \epsilon. \qquad (2)$$

*Proof.* We first note that we use a different constant $C_2 = 2C_1$ in the theorem as compared to lemma 1 which helps ensure that,

$$C_2 n_i \log \left( \frac{n_i n_{i+1} \max(|\mathcal{B}_{i \to i+1}|, \||\mathcal{B}_{i \to i+1}\||) \, l}{\min(\epsilon, \delta)} \right) \geq C_1 n_i \log \left( \frac{2 n_i n_{i+1} \max(|\mathcal{B}_{i \to i+1}|, \||\mathcal{B}_{i \to i+1}\||) \, l}{\min(\epsilon, \delta)} \right),$$

which is true in the domain of the following variables $(n_i, n_{i+1}, l \geq 1, \delta \leq, \epsilon \leq \frac{1}{2}, |\mathcal{B}_{i \to i+1}[\geq 1)$ as $2 \log(x) \geq \log(2x)$ on $[2, +\infty]$.

We first apply lemma 1 $l$-times for each layer of the target network with $\epsilon$ becoming $\frac{\epsilon}{2l}$ and $\delta$ becoming $\frac{\delta}{l}$. With an overparametrization factor $\tilde{n}_i \geq C_1 n_i \log(\frac{2 n_i n_{i+1} \max(|\mathcal{B}_{i \to i+1}|, \||\mathcal{B}_{i \to i+1}\||)l}{\min(\epsilon, \delta)})$ we get that for each layer $i$,

$$\max_{\|\mathbf{x}\| \leq 1} \|(\mathbf{S}_{2i+1} \odot \mathbf{W}_{2i+1}^h)\sigma((\mathbf{S}_{2i} \odot \mathbf{W}_{2i}^h)\mathbf{x}) - f_i(\mathbf{x})\| \leq \frac{\epsilon}{2l} \qquad (15)$$

holds with probability at least $1 - \frac{\delta}{l}$. By taking a union bound, we get that this holds for every layer with probability at least $1 - \delta$. Now, let $x_i'$ be the input to the $(2i)$-th layer of the pruned overparametrized network $h$. Furthermore, let $x_i$ be the input to the $i$-th layer of the target network $f$. Then we have,

- $x_0' = x_0 = x$

- $x_{i+1}' = \sigma\left((\mathbf{S}_{2i+1} \odot \mathbf{W}_{2i+1}^h)\sigma\left((\mathbf{S}_{2i} \odot \mathbf{W}_{2i}^h)\mathbf{x}_i'\right)\right)$ for $i \leq l-2$

- $x_l' = (\mathbf{S}_{2l-1} \odot \mathbf{W}_{2l-1}^h)\sigma\left((\mathbf{S}_{2l-2} \odot \mathbf{W}_{2l-2}^h)\mathbf{x}_{l-1}'\right)$

Equation 15 implies that,

$$\|(\mathbf{S}_{2i+1} \odot \mathbf{W}_{2i+1}^h)\sigma((\mathbf{S}_{2i} \odot \mathbf{W}_{2i}^h)\mathbf{x}_i') - f_i(\mathbf{x}_i')\| \leq \|\mathbf{x}_i'\| \frac{\epsilon}{2l} \qquad (16)$$

Passing through the point-wise ReLU which is 1-Lipschitz for all the norms that we work with we get:

$$\|\mathbf{x}_{i+1}'\| \leq \|\mathbf{x}_i'\| \left(1 + \frac{\epsilon}{2l}\right) \qquad (17)$$

By leveraging a recursive argument, and using the fact that $\|\mathbf{x}_0'\| = \|\mathbf{x}\| \leq 1$ we then get that for all $i \in \{0, \ldots, l-1\}$, $\quad \|x_i'\| \leq (1 + \frac{\epsilon}{2l})^i$. Then, forall $i \leq l-2$:

$$\begin{aligned}
\|\mathbf{x}_{i+1}' - \mathbf{x}_{i+1}\| &= \|\sigma\left((\mathbf{S}_{2i+1} \odot \mathbf{W}_{2i+1}^h)\sigma((\mathbf{S}_{2i} \odot \mathbf{W}_{2i}^h)\mathbf{x}_i')\right) - \sigma\left(f_i(\mathbf{x_i})\right)\| \\
&\leq \|\sigma\left((\mathbf{S}_{2i+1} \odot \mathbf{W}_{2i+1}^h)\sigma((\mathbf{S}_{2i} \odot \mathbf{W}_{2i}^h)\mathbf{x}_i')\right) - \sigma\left(f_i(\mathbf{x}_i')\right)\| + \|\sigma\left(f_i(\mathbf{x}_i')\right) - \sigma\left(f_i(\mathbf{x_i})\right)\| \\
&\leq \|\mathbf{x}_i'\| \frac{\epsilon}{2l} + \||f_i\||\|\mathbf{x}_i' - \mathbf{x}_i\| \\
&\leq \left(1 + \frac{\epsilon}{2l}\right)^i \frac{\epsilon}{2l} + \|\mathbf{x}_i' - \mathbf{x}_i\|,
\end{aligned}$$

where we used the fact that $\sigma$ is one Lipschitz. We then get that,

$$\|x'_l - x_l\| = \|(\mathbf{S}_{2l-1} \odot \mathbf{W}^h_{2l-1})\sigma\left((\mathbf{S}_{2l} \odot \mathbf{W}^h_{2l})\mathbf{x}'_{l-1}\right)) - f_{l-1}(\mathbf{x}_{l-1})\|$$

$$\leq \|(\mathbf{S}_{2l-1} \odot \mathbf{W}^h_{2l-1})\sigma((\mathbf{S}_{2l-2} \odot \mathbf{W}^h_{2l-2})\mathbf{x}'_{l-1}) - f_{l-1}(\mathbf{x}'_{l-1})\| + \|f_{l-1}(\mathbf{x}'_{l-1}) - f_{l-1}(\mathbf{x}_{l-1})\|$$

$$\leq \|\mathbf{x}'_{l-1}\|\frac{\epsilon}{2l} + \|f_{l-1}\|\|\mathbf{x}'_{l-1} - \mathbf{x}_{l-1}\|$$

$$\leq (1 + \frac{\epsilon}{2l})^{l-1}\frac{\epsilon}{2l} + \|\mathbf{x}'_{l-1} - \mathbf{x}_{l-1}\|$$

$$\leq \sum_{i=0}^{l-1}\left(1 + \frac{\epsilon}{2l}\right)^i \frac{\epsilon}{2l}$$

$$\leq (1 + \frac{\epsilon}{2l})^l - 1$$

$$\leq e^{\frac{\epsilon}{2}} - 1$$

$$\leq \epsilon \quad \text{because } \epsilon \leq \frac{1}{2}$$

$\square$

## B.3 LOWER BOUND

We now prove Theorem 2.

**Theorem 2.** *Let $\hat{h}_i$ be a network with $\Theta$ parameters such that:*

$$\forall f_i \in \mathcal{F}_i, \exists S_i \in \{0,1\}^\Theta \text{ such that } \max_{\mathbf{x} \in \mathbb{R}^{D_i \times n_i}, \|\mathbf{x}\| \leq 1} \|(S_i \odot \hat{h}_i)(\mathbf{x}) - f_i(\mathbf{x})\| \leq \epsilon. \quad (3)$$

*Then $\Theta$ is at least $\Omega\left(n_i n_{i+1}|\mathcal{B}_{i \to i+1}|\log(\frac{1}{\epsilon})\right)$ and $\tilde{n}_i$ is at least $\Omega(n_i \log\left(\frac{1}{\epsilon}\right))$ in Theorem 1.*

Let us first recall the main assumptions of our setting:

- For all $f_i \in \text{Span}(\kappa_{n_i \to n_{i+1}} \otimes \mathcal{B}_{i \to i+1})$, where $f_i = \sum_{p \in [n_i]} \sum_{q \in [n_{i+1}]} \sum_k \alpha^{(i)}_{p \to q,k} b_{i \to i+1,k}$, we have: $\|f_i\| \leq 1 \implies \|\alpha^{(i)}\|_\infty \leq 1$. This assumption is extremely mild as it can always trivially be satisfied by rescaling the basis elements.
- $\exists M_1 \in \mathbb{R}^+$ such that, uniformly over $i$, for every possible building block in our equivariant feature spaces $\mathbb{F}_i$ and $\mathbb{F}_{i+1}$ we have, $|\mathcal{B}_{i \to i}| \leq M_1|\mathcal{B}_{i \to i+1}|$. This assumption is used to mainly guard against a non-trivial scenario where the first layer would be able to carry "a lot of superfluous parameters". This assumption finds its solitary use in achieving the lower bound on the over-parametrization factor in the theorem 1, i.e. to prove $\tilde{n}_i \geq \Omega(n_i \log\left(\frac{1}{\epsilon}\right))$ and is not used for the lower bound on $\Theta$. This assumption is very mild because in most of the usual cases (MLPs, CNNs, E(2)-steerable CNNs) the possible building blocks of each layer $\mathbb{F}_i$ are finite (respectively $\mathbb{R}$, $\mathbb{R}^{d^2}$ and $\mathbb{R}^{d^2}$ or $\mathbb{R}^{d^2 \times |G|}$). Being finite automatically implies the existence of such a constant $M_1$ as we can simply take the maximum over the possible values of $\frac{|\mathcal{B}_{i \to i+1}|}{|\mathcal{B}_{i \to i}|}$.
- Finally we assume the existence of a constant $M_2 \in \mathbb{R}^+$ such that $n_i \leq M_2 n_{i+1}$. This mild assumption—like the previous one—is used to ensure that $\tilde{n}_i \geq \Omega(n_i \log\left(\frac{1}{\epsilon}\right))$.

Our proof relies on a counting based argument that compares the number of pruning masks to the cardinal of a $2\epsilon$-separated net $\mathcal{P}$ in the set of target networks with respect to the operator norm. A similar argument was used by Pensia et al. (2020) in the context of dense nets. We recall here the definition of a $2\epsilon$-separated net:

**Definition B.1.** *Let $F$ be a normed vector space. A $2\epsilon$-separated net $\mathcal{P}$, in $F$ is a subset $\mathcal{P} \subset F$ such that:*

$$\forall x_1, x_2 \in \mathcal{P}, \quad x_1 \neq x_2 \implies \|x_1 - x_2\| \geq 2\epsilon$$

In Lemma 1, we considered only a set of target network $\mathcal{F}_i \subset \text{Span}(\kappa_{n_i \to n_{i+1}} \otimes \mathcal{B}_{i \to i+1})$ where each function has $\|f_i\| \leq 1$, and $\|\alpha^{(i)}\|_\infty \leq 1$. Mixing this with the first assumption written

above, it is therefore the set of maps $f_i$ such that $\|\alpha^{(i)}\|_\infty \leq 1$. Now consider the isomorphism $\mathcal{I}_i : \mathrm{Span}(\kappa_{n_i \to n_{i+1}} \otimes \mathcal{B}_{i \to i+1}) \to \mathbb{R}^{n_i n_{i+1} |\mathcal{B}_{i \to i+1}|}$, which identify a function with its coefficients in the equivariant basis.

$$\mathcal{I}_i := f_i \mapsto \alpha^{(i)} \tag{18}$$

This isomorphism shows that $\mathcal{F}_i$ can be seen as the norm ball of $\mathbb{R}^{n_i n_{i+1} |\mathcal{B}_{i \to i+1}|}$ with respect to the norm induced on $\mathbb{R}^{n_i n_{i+1} |\mathcal{B}_{i \to i+1}|}$ by the isomorphism. Moreover $\mathcal{P}$ is a $2\epsilon$-separated net on $\mathcal{F}_i$ if and only if its image is a $2\epsilon$-separated net on $\mathbb{R}^{n_i n_{i+1} |\mathcal{B}_{i \to i+1}|}$ with respect to the induced norm.

**Lower bound on $|\mathcal{P}|$.** We just need to use Lemma 4.2.8 and extend Proposition 4.2.12 from Vershynin (2018) to non-Euclidean balls. The general idea is as follows: Denote by $\mathcal{B}(x, R)$ the ball centered at $x$ of radius $R$ in $\mathbb{R}^{n_i n_{i+1} |\mathcal{B}_{i \to i+1}|}$ and by $\mu$ the Lebesgue measure. Let us construct a $2\epsilon$-separated net as follow: we take for first point the origin $0$ of the vector space. At the step $n$, to construct the $n+1$-th point of $\mathcal{P}$ we proceed as follow: if $\mathcal{B}(0, 1) \subset \bigcup_{x \in \mathcal{P}} \mathcal{B}(x, 2\epsilon)$ we stop the processus and don't take any $n+1$-th point. Else, we take a point in $\mathcal{B}(0, 1) \setminus \bigcup_{x \in \mathcal{P}} \mathcal{B}(x, 2\epsilon)$. We know that this point is at a distance of at least $2\epsilon$ of the other points of $\mathcal{P}$. Moreover it is in the unit ball. At the end of the process (which must end since the unit ball is compact), we finally get that for the $2\epsilon$-separated net $\mathcal{P}$, $\mathcal{B}(0, 1) \subset \bigcup_{x \in \mathcal{P}} \mathcal{B}(x, 2\epsilon)$. Therefore, $\mu(\mathcal{B}(0, 1)) \leq \mu(\bigcup_{x \in \mathcal{P}} \mathcal{B}(x, 2\epsilon)) \leq |\mathcal{P}| \mu(\mathcal{B}(0, 2\epsilon))$. Finally, $|\mathcal{P}| \geq \frac{\mu(\mathcal{B}(0,1))}{\mu(\mathcal{B}(0,2\epsilon))} = \left(\frac{1}{2\epsilon}\right)^{n_i n_{i+1} |\mathcal{B}_{i \to i+1}|}$. In the last step, we use the fact that the Lebesgue measure of a ball of radius $R$ in a vector space of dimension $n$ is $R^n V_n$ where $V_n$ is the Lebesgue measure of the unit ball. This allows to choose $|\mathcal{P}| \geq \left(\frac{1}{2\epsilon}\right)^{n_i n_{i+1} |\mathcal{B}_{i \to i+1}|}$.

**Lower bound induced on $\Theta$.** As the network that we seek to prune has $\Theta$ parameters, the number of binary pruning masks that can be constructed is $2^\Theta$. Moreover, due to the triangular inequality, each pruned network can approximate at most one element of $\mathcal{P}$. Indeed, if $f_i^1 \neq f_i^2 \in \mathcal{P}$ are approximated with the same pruning mask,

$$\|f_i^2 - f_i^1\| \leq \|f_i^2 - (S \odot \hat{h}_i)\| + \|(S \odot \hat{h}_i) - f_i^1\| \leq 2\epsilon, \tag{19}$$

which contradicts the fact that $\mathcal{P}$ is a $2\epsilon$-separated net. This directly implies that the number of pruning masks must be bigger than the cardinal of $\mathcal{P}$.

$$2^\Theta \geq \left(\frac{1}{2\epsilon}\right)^{n_i n_{i+1} |\mathcal{B}_{i \to i+1}|} \tag{20}$$

and by taking the $\log$,

$$\Theta \geq \frac{n_i n_{i+1} |\mathcal{B}_{i \to i+1}|}{\log(2)} \log\left(\frac{1}{2\epsilon}\right) \tag{21}$$

which shows that $\Theta$ must be at least $\Omega(n_i n_{i+1} |\mathcal{B}_{i \to i+1}| \log\left(\frac{1}{\epsilon}\right))$

**Lower bound on $\tilde{n}_i$.** We now seek to provide a lower bound on $\tilde{n}_i$ such that Theorem 1 holds. Since, our main claim requires that we approximate every target network with probability at least $1 - \delta > 0$, the set of parameters (drawn from any distribution) that can achieve this is non zero. What remains is to count the number of parameters contained within the overparametrized $G$-equivariant network in $\mathcal{H}_i$ (see Lemma 1) as a function of the overparametrization factor $\tilde{n}_i$. This allows us to lower bound $\tilde{n}_i$ via the lower bound on the number of parameters established above. Now any overparametrized $G$-equivariant network we construct has the following number of parameters:

- Number of parameters of the first layer: $n_i \tilde{n}_i |\mathcal{B}_{i \to i}|$
- Number of parameters of the second layer: $\tilde{n}_i n_{i+1} |\mathcal{B}_{i \to i+1}|$

Therefore the overparametrized network $h_i$ has $\Theta = \tilde{n}_i(n_i |\mathcal{B}_{i \to i}| + n_{i+1} |\mathcal{B}_{i \to i+1}|)$ parameters. Using the second and third assumptions, we get that:

$$\Theta \leq \tilde{n}_i(M_1 M_2 n_{i+1} |\mathcal{B}_{i \to i+1}| + n_{i+1} |\mathcal{B}_{i \to i+1}|) \leq \tilde{n}_i(M_1 M_2 + 1)n_{i+1} |\mathcal{B}_{i \to i+1}|. \tag{22}$$

Moreover by Eq. 21, we know that:

$$\Theta \geq \Omega\left(n_i n_{i+1}|\mathcal{B}_{i\to i+1}|\log\left(\frac{1}{\epsilon}\right)\right).$$

It therefore implies that:

$$\tilde{n}_i \geq \Omega\left(n_i \log\left(\frac{1}{\epsilon}\right)\right) \tag{23}$$

**Discussion.** Using the result in Theorem 2 we can now understand that Theorem 1 informs us that our proposed overparametrization strategy is optimal with respect to the tolerance $\epsilon$ and almost optimal with respect to $n_i n_{i+1}|\mathcal{B}_{i\to i+1}|$. In Theorem 1 we observe an additional factor of $\log(n_i n_{i+1}\max(|\mathcal{B}_{i\to i+1}|, \|\mathcal{B}_{i\to i+1})\|)$ which appears in $\tilde{n}_i$. We can reconcile this term which appears in the proof due to both our choice of with which metric do we want to approximate the target network and to the probabilistic setting of the SLTH.

Indeed, first we note that we chose to approximate each target layer by $\epsilon$ with respect to the operator norm associated with the norms on the input and output space. But such a choice is arbitrary, and if we had chosen another metric, such as approximating each weight of the target network in the diamond shape structure by $\epsilon$, then the term $n_i n_{i+1}\|\mathcal{B}_{i\to i+1}\|$ might have been eliminated.

The term $n_i n_{i+1}|\mathcal{B}_{i\to i+1}|$ arises from the fact that the parameters of the overparametrized network are drawn from a random process. Specifically, a bigger overparametrization is needed because of the scenario when not all the SUBSET-SUM problems have solutions, which has a probability of occurring that grows with $n_i n_{i+1}|\mathcal{B}_{i\to i+1}|$—i.e. the complexity of the approximation. We could replace the probabilistic setting by instead taking an overparametrized network deterministically initilized by a smart initialization such that with probability one all possible subnetworks can be obtained by pruning the overparametrized one. In this case, the overparametrization on the width would no longer have the term $n_i n_{i+1}|\mathcal{B}_{i\to i+1}|$ in the log. Such an initialization can be taken for example by decomposing the overparametrized network in the different blocks of the diamond shape and taking the weights in each block to be $\pm 1, \pm\frac{1}{2}, \pm\frac{1}{4}, \pm(\frac{1}{2})^{\frac{\log(\epsilon)}{\log(2)}} = \frac{1}{\epsilon}$ (the weights that are not part of a diamond shape can be initialized freely). Each weight of the target network can then be approximated by pruning the diamond shape with a mask which is the binary writing of the target weight. This is possible for every weight of the target network and for all target network at once with probability one (with a different mask for each target network). We note here the similarity of this construction with the one used in Sreenivasan et al. (2022), albeit under a different setting than the one we considered here.

In conclusion, we give some hints to annihilate the term $n_i n_{i+1}\max(|\mathcal{B}_{i\to i+1}|, \|\mathcal{B}_{i\to i+1}\|)$: first choosing another metric for approximating a layer and secondly going to a non-probabilistic setting where the overparametrized network is smartly initialized.

## C  PROOF OF STL ON MLP USING THEOREM 1

This corollary recovers the main result of Pensia et al. (2020).

In this case, $G = \{e\}$ and the representation is trivial. The building block of a layer is $\mathbb{F}_i = \mathbb{R}$ and each layer is composed of a stack of $n_i$, i.e. $\mathbb{R}^{n_i} = \mathbb{F}_i^{n_i}$. The norm that we will use on $\mathbb{R}$ is of course the absolute value $|\cdot|$. Therefore, as explained above, the norm that we consider on $\mathbb{F}_i^{n_i} = \mathbb{R}^{n_i}$ is $\|\cdot\|_\infty$. The pointwise ReLU is trivially equivariant, since the $G$ is trivial. It is moreover 1-Lipschitz.

All maps are equivariant, since the group $G$ is trivial. An equivariant basis of the maps $\mathbb{F}_i \to \mathbb{F}_{i+1}$ and of the maps $\mathbb{F}_i \to \mathbb{F}_i$ is therefore a basis of the maps $\mathbb{R} \to \mathbb{R}$. It is of dimension 1 and of course taken to be the identity. We therefore obtain that the identity is in $\mathcal{B}_{i\to i}$. One has $|\mathcal{B}_{i\to i}| = 1$ and $\|\mathcal{B}_{i\to i+1}\| = \max_{|\alpha|\leq 1}\|\alpha\mathbb{I}\| = 1$.

All the conditions are therefore validated and we are free to apply Theorem 1 in this setting, with $\max(|\mathcal{B}_{i\to i}|, \|\mathcal{B}_{i\to i+1}\|) = 1$ which leads to the following corollary:

**Corollary 3.** *Let $h \in \mathcal{H}$ be a random MLP of depth $2l$, i.e., $h(\mathbf{x}) = \mathbf{W}_{2l-1}^h \sigma \left( \dots \sigma(\mathbf{W}_0^h x) \right)$ where $\mathbf{W}_{2i}^h \in \mathbb{R}^{n_i \times \tilde{n}_i}$, $\mathbf{W}_{2i+1}^h \in \mathbb{R}^{\tilde{n}_i \times n_{i+1}}$ are dense linear maps with weights drawn from $\mathcal{U}([-1,1])$ If $\tilde{n}_i = C_2 n_i \log\left( \frac{n_i n_{i+1} l}{\min(\epsilon, \delta)} \right)$, then with probability at least $1 - \delta$ we have that for all $f \in \mathcal{F}$ a target MLP with layers $\mathbf{W}_i^f \in [-1,1]^{n_i \times n_{i+1}}$ and $\|\|f_i\|\| \le 1$ there exists a collection of pruning masks $\mathbf{S}_{2l-1}, \dots, \mathbf{S}_0$ such that,*

$$\max_{\mathbf{x} \in \mathbb{R}^{n_0}, \|\mathbf{x}\| \le 1} \|(\mathbf{S}_{2l-1} \odot \mathbf{W}_{2l-1}^h)\sigma\left( \dots \sigma\left( \left(\mathbf{S}_0 \odot \mathbf{W}_0^h\right) x\right)\right) - f(x)\| \le \epsilon \tag{24}$$

## D  PROOF OF SLT ON CNN USING THEOREM 1

We now prove Theorem 1 application to the case regular translation equivariant CNNs. We highlight here that this is a strict generalization of the result obtained by da Cunha et al. (2022b) as we do not assume strictly positive inputs (recently extended in parallel in Burkholz (2022a)).

**Corollary 4.** *Let $h \in \mathcal{H}$ be a random CNN of depth $2l$, i.e., $h(\mathbf{x}) = \mathbf{K}_{2l-1}^h * \sigma \left( \dots \sigma(\mathbf{K}_0^h * x) \right)$ where $\mathbf{K}_{2i}^h \in \mathbb{R}^{d^2 \times n_i \times \tilde{n}_i}$, $\mathbf{K}_{2i+1}^h \in \mathbb{R}^{d^2 \times \tilde{n}_i \times n_{i+1}}$ are convolutional kernels with weights in $\mathcal{U}([-1,1])$ If $\tilde{n}_i = C_2 n_i \log\left( \frac{d^2 n_i n_{i+1} l}{\min(\epsilon, \delta)} \right)$, then with probability at least $1 - \delta$ we have that for all $f \in \mathcal{F}$ a target CNN with kernels $\mathbf{K}_i^f \in [-1,1]^{d^2 \times n_i \times n_{i+1}}$ and $\|\|f_i\|\| \le 1$ there exists a collection of pruning masks $\mathbf{S}_{2l-1}, \dots, \mathbf{S}_0$ such that,*

$$\max_{\mathbf{x} \in \mathbb{R}^{d^2 \times n_0}, \|\mathbf{x}\| \le 1} \|(\mathbf{S}_{2l-1} \odot \mathbf{K}_{2l-1}^h) * \sigma\left( \dots \sigma\left( \left(\mathbf{S}_0 \odot \mathbf{K}_0^h\right) * x\right)\right) - f(x)\| \le \epsilon \tag{25}$$

We now prove Corollary 4. In our case, the building blocks of every layer are $\mathbb{F}_i = \mathbb{R}^{d^2}$ where $d^2$ is the size of an image. Therefore, $\mathcal{B}_{i \to i+1}$ is the basis of translation equivariant maps: $\mathbb{R}^{d^2} \to \mathbb{R}^{d^2}$. When working with CNNs, the basis that is used in practice is the convolution with a kernel $\mathbf{K}^{p,q} \in \mathbb{R}^{d^2}$ where $\mathbf{K}^{p,q}$ has only a 1 at the index $(p, q)$ and is filled everywhere else with zeros on the grid $d \times d$, where $(p, q) \in [d]^2$. It is therefore easy to see that :

$$|\mathcal{B}_{i \to i+1}| = d^2.$$

Let us choose $\| \cdot \|_\infty$ as a norm on $\mathbb{R}^{d^2}$. Applying the proposition 1 from da Cunha et al. (2022b), we get:

$$\forall \mathbf{K} \in \mathbb{R}^{d \times d}, \quad \forall \mathbf{X} \in \mathbb{R}^{d \times d}, \quad \|\mathbf{K} * \mathbf{X}\|_\infty \le \|\mathbf{K}\|_1 \|\mathbf{X}\|_\infty.$$

By using this basis we then get that,

$$\begin{aligned} \|\|\mathcal{B}_{i \to i+1}\|\| &= \max_{\mathbf{K} \in [-1,1]^{d^2}} \max_{\mathbf{X} \in [-1,1]^{d^2}} \|\mathbf{K} * \mathbf{X}\|_\infty \\ &\le \max_{\mathbf{K} \in [-1,1]^{d^2}} \max_{\mathbf{X} \in [-1,1]^{d^2}} \|\mathbf{K}\|_1 \|\mathbf{X}\|_\infty \\ &\le d^2. \end{aligned}$$

We then get that:

$$\max(|\mathcal{B}_{i \to i+1}|, \|\|\mathcal{B}_{i \to i+1}\|\|) = d^2.$$

It is trivial to notice that the pointwise-ReLU used is equivariant and 1-Lipschitz. Moreover, the identity is clearly in $\mathcal{B}_{i \to i}$ by taking the kernel with only a 1 at the origin. Therefore all the conditions

are met and we can apply theorem 1 which states that the overparametrization needed is:

$$\tilde{n}_i = C_2 n_i \log \left( \frac{n_i n_{i+1} \max \left( |\mathcal{B}_{i \to i+1}|, \|\|\mathcal{B}_{i \to i+1}\|\| \right) l}{\min \left( \epsilon, \delta \right)} \right)$$

$$= C_2 n_i \log \left( \frac{d^2 n_i n_{i+1} l}{\min \left( \epsilon, \delta \right)} \right).$$

# E   ADDITIONAL MATERIAL ON $E(2)$-STEERABLE NETWORKS

## E.1   GENERAL EQUIVARIANT LAYERS IN THE CASE OF FEATURE FIELDS DEFINED ON $\mathbb{R}^2$

In full generality, the theory of $E(2)$-steerable CNN has been developed in the setting of continuous and infinite steerable fields defined on $\mathbb{R}^2$. The input and output of a layer are then respectively functions in $(\mathbb{R}^2 \to \mathbb{R}^{c_{\text{in}}})$ and in $(\mathbb{R}^2 \to \mathbb{R}^{c_{\text{out}}})$. The reader will immediately note that it does not correspond to the practical case of $E(2)$-steerable CNN since these type of inputs are not infinite dimensional. The condition for a layer to be equivariant between these two feature fields is to be written as a continuous convolution with kernels satisfying the condition (called equivariant kernels):

$$\pi(g \cdot x) = \rho_{\text{out}}(g)\pi(x)\rho_{\text{in}}^{-1}(g), \quad \forall g \in G, x \in \mathcal{X} \tag{26}$$

There are different methods to compute the possible kernels that satisfy this condition, that will lead to different basis. For example, in Weiler and Cesa (2019), the authors use the polar coordinates to solve this condition. They have a free parameter which is the frequency and by varying this parameter they can compute a basis of the equivariant kernels.

Our method to construct a basis of the equivariant kernels is different: we quotient the plane $\mathbb{R}^2$ by the equivalence relation induced by the orbits under the group $G$. For each point in the continuous quotient space $\mathcal{A}_{\mathcal{R}}$, we compute a basis of the equivariant kernels by putting an element of the canonical basis at this point and summing over the group $G$ the action of an element of $G$ on this element. More precisely, we impose having some matrix $\mathbf{K}_{0,x}^{p,q}$ at the point $x \neq 0$ and to obtain the full equivariant kernel, we just apply the following formula:

$$\forall y \in \mathbb{R}^2 \quad b(y) := \mathbf{K}_{G,x}^{p,q}(y) = \sum_{g \in G} \rho_{i+1}(g)\mathbf{K}_{0,x}^{p,q}(g^{-1}y)\rho_i(g^{-1}). \tag{27}$$

This formula is well defined because in the case of subgroups of $O(2)$, $\forall x, y \in \mathbb{R}^2 \backslash \{0\}$, the set $\{g \in G, g \cdot x = y\}$ is finite meaning that the above sum is finite for every $y \in \mathbb{R}^2$. It remains to check that the kernel $\mathbf{K}_{G,x}^{p,q}$ respects the above condition on equivariant kernels. Indeed, one has that:

$$\forall y \in \mathbb{R}^2, \forall h \in G, \quad \mathbf{K}_{G,x}^{p,q}(h \cdot y) = \sum_{g \in G} \rho_{i+1}(g)\mathbf{K}_{0,x}^{p,q}(g^{-1} \cdot (h \cdot y))\rho_i(g^{-1})$$

$$= \sum_{g \in G} \rho_{i+1}(h)\rho_{i+1}(h^{-1}g)\mathbf{K}_{0,x}^{p,q}((h^{-1}g)^{-1} \cdot y))\rho_i(g^{-1}h)\rho_i(h)^{-1}$$

$$= \rho_{i+1}(h) \left( \sum_{g \in G} \rho_{i+1}(h^{-1}g)\mathbf{K}_{0,x}^{p,q}((h^{-1}g)^{-1} \cdot y))\rho_i(g^{-1}h) \right) \rho_i(h)^{-1}$$

$$= \rho_{i+1}(h)\mathbf{K}_{G,x}^{p,q}(y)\rho_i(h)^{-1}$$

where we used that $g \mapsto h^{-1}g$ from $G$ to $G$ is a bijection. One should note that for some groups $G$ and some $x \neq 0$ it may be possible that $\exists g \in G, \quad g \cdot x = x$. The set of all elements that keep the point unchanged is known as the stabilizer subgroup. For example, for $G$ a dihedral group and a point $x$ on the symmetry axis, one has that $x$ remains untouched by the symmetry with respect to this axis. This is however not a problem, as the set of such $g$ is finite, and therefore the above formula is still valid, even at the point $x$. One will note however that $\mathbf{K}_{G,x}^{p,q}(x) \neq \mathbf{K}_{0,x}^{p,q}(x)$. This means that we may lose the fact that the set of equivariant kernels $\{\mathbf{K}_{G,x}^{p,q}, (p,q) \in [c_{\text{in}}] \times [c_{\text{out}}]\}$ is composed of independent vectors and therefore forms a basis. We will still have that it spans the space of equivariant kernels but not that it will form a basis.

## E.2   CONSTRUCTION OF THE KERNEL AT THE ORIGIN

We would like to apply our basis construction formula to every point in the plane, including the origin but the problem is that at the origin: $\forall g \in G, \quad g \cdot 0 = 0$. Therefore the above sum is not well defined because it is infinite for infinite groups. We can only apply this formula in the case of $G$ finite. The usual way to solve the problem at the origin if one deals with infinite groups is to solve all the linear problems $\pi(g \cdot x) = \rho_{\text{out}}(g)\pi(x)\rho_{\text{in}}(g)^{-1}$. However in the setting of Corollary 3, we deal with finite subgroups of O(2). Therefore we can apply the above formula:

$$\forall y \in \mathbb{R}^2, \quad b(y) := \mathbf{K}_{G,0}^{p;q}(y) = \sum_{g \in G} \rho_{i+1}(g)\mathbf{K}_{0,0}^{p;q}(g^{-1}y)\rho_i(g^{-1}).$$

In our case, when dealing with the regular representation, if one takes $G = C_n$ the cyclic group of n rotations, one will check that the $\rho_i(g)$ are permutation matrices associated with the permutation of $G : h \mapsto g \cdot h$. One can then check that summing over $G$ leads to a circulant matrix.

We have thus computed the set of equivariant kernels at the origin by using the above formula. One may have wanted to solve all the linear problems set by the equivariant constraints. Here they can be reformulated by the fact that the kernel at the origin must commute with all the matrices associated with the permutations of $G : h \mapsto g \cdot h$. Solving this leads to the set of circulant matrices.

## E.3   DISCRETIZATION OF $\mathbb{R}^2$

We now highlight the practical challenges of building equivariant networks and their associated pruning when we discretize continuous signals on $\mathbb{R}^2$ to a pixelized grid.

**From $\mathbb{R}^2$ to $[-\frac{d}{2}, \frac{d}{2}]^2$.** The first problem we want to address is that we do not usually work on the plane $\mathbb{R}^2$ but on spatially delimited images on $[-\frac{d}{2}, \frac{d}{2}]^2$. This is problematic since when $G$ acts on a square images, it can become a non-square image after a rotation. For example, $C_8$ doe not always send $[-\frac{d}{2}, \frac{d}{2}]^2$ on $[-\frac{d}{2}, \frac{d}{2}]^2$ (take the rotation by 45° for instance). In the same way restricting the equivariant kernel to a finite space $[-\frac{d}{2}, \frac{d}{2}]^2$ as it is done in usual CNNs would lead to problems since for some $x \in [-\frac{d}{2}, \frac{d}{2}]^2$, and some $g \in G, g \cdot x \notin [-\frac{d}{2}, \frac{d}{2}]^2$. We overcome this issue by restricting the kernels to not being defined on $\mathbb{R}^2$ but on a disk centered at the origin whose diameter equals the size of the image (see Figure 2). To implement this, we multiply with a mask which exponentially decays to zero for points with radius larger than the radius of the disk. This is permitted because the equivariant constraint set constrains the interior of the orbit, and it is trivial that for sub-groups of O(2), the disc is stable under the action of the group. Therefore, the kernels that we obtain are still equivariant because they can check the equivariant constraint.

**From $[-\frac{d}{2}, \frac{d}{2}]^2$ to $\{-\frac{d}{2}, -\frac{d-2}{2}, ..., \frac{d-2}{2}, \frac{d}{2}\}^2$.** The second problem that we must address is the discretization process. Indeed, we do not work with continuous feature fields $f : [-\frac{d}{2}, \frac{d}{2}]^2 \to \mathbb{R}^c$ but with pixellized images, i.e. discretized inputs $f : \{-\frac{d}{2}, -\frac{d-2}{2}, ..., \frac{d}{2}\}^2 \to \mathbb{R}^c$. This a problem, because the equivariant constraint Eq. 26 puts constraints between $k(g \cdot x)$ and $k(x)$. Equation 26 cannot be used anymore because $g \cdot x$ is not always on the grid. For instance, if $x = (1, 1)$ and $g$ is the rotation by 45°, then $g \cdot x = (0, \sqrt{2}) \notin \{-\frac{d}{2}, -\frac{d-2}{2}, ..., \frac{d}{2}\}^2$. Moreover, note that it is not sufficient to discretize the equivariant kernels: one must choose only a finite subset of them. Indeed, the dimension of the equivariant map must be finite in the discretized setting as opposed to the continuous setting where it is infinite. In practice, the network is not exactly equivariant, but almost equivariant due to a discretization error. However, this is not an issue in the setting of Theorem 1. Indeed, once we have chosen a basis of the "almost-equivariant" kernels, we can prove the SLTH for the class of such networks, which is exactly the result that we want in practice.

Weiler and Cesa (2019) choose a finite subset of the equivariant kernels, the authors upper-bound the frequency of the polar coordinate solution by an anti-aliasing condition. They then discretize the continuous kernels on the grid. For our basis construction, we choose a finite subset of the equivariant kernels by restricting $\mathcal{A}_\mathcal{R}$ to only $\mathcal{A}_\mathcal{R} \bigcap \{-\frac{d}{2}, -\frac{d-2}{2}, ..., \frac{d}{2}\}^2$. There are many different ways to discretize our kernels $\mathbf{K}_{G,x}^{p;q}$ defined on $\mathbb{R}^2$. One way would be to send $g \cdot x$ to the nearest pixel if it is not on the grid. In order to decrease the discretization error, we first upsample the grid by a factor 3 before we start applying actions of the group $G$ to the base space. For the subgroups of

O(2) we consider, rotations of the base space are performed using bilinear interpolation. Finally, we downsample to the original size in order to obtain the discretized version of $\mathbf{K}_{G,x}^{p,q}$.

### E.4    PROOF OF SLT ON E(2)-STEERABLE CNNs USING THEOREM 1

We now prove Corollary 1. We work with trivial or regular representations of $G \leq \mathrm{O}(2)$ on top of feature fields. It is straightforward that the pointwise ReLU is equivariant. Moreover, to easily compute $\|\|\mathcal{B}_{i \to i+1}\|\|$ we work with $\| \cdot \|_\infty$ which implies that the ReLU is then 1-Lipschitz. Finally, the identity can trivially be written as the convolution with an equivariant kernel having the identity at the origin (the identity is trivially a circulant matrix). Therefore we have that $\mathbb{I} \in \mathcal{B}_{i \to i}$

If we use a trivial representation on top of the feature field at layer $i$, then the building block of this layer is $\mathbb{F}_i = \mathbb{R}^{d^2}$. If we instead use a regular representation, then the building block of this layer is $\mathbb{F}_i = \mathbb{R}^{d^2 \times |G|}$. From the construction of the equivariant basis, we deduce that $|\mathcal{B}_{i \to i+1}| \leq d^2 |G|^2$ for each layer. Indeed, we must first choose a pixel on the set of representatives $\mathcal{A}_{\mathcal{R}} \subset \{-\frac{d}{2}, -\frac{d-2}{2}, ..., \frac{d-2}{2}, \frac{d}{2}\}^2 \simeq [d] \times [d]$ grid, and then choose a subset of the canonical basis at this point. But such canonical basis has $|G| \times |G|$ elements for regular to regular, $1 \times |G|$ element for trivial to regular (or regular to trivial), and finally only $1 \times 1$ for trivial to trivial. This is even less than that at some points such as the origin because of the additional constraints. Finally, one has less that $d^2 \times |G|^2$ choices in all cases which indicates that,

$$|\mathcal{B}_{i \to i+1}| \leq d^2 |G|^2.$$

In fact, since we can only choose $x \in \mathcal{A}_{\mathcal{R}}$ to obtain a set of independent elements, the true dependency will be $|\mathcal{B}_{i \to i+1}| \simeq |\mathcal{A}_{\mathcal{R}}| \cdot |G|^2 \simeq \frac{d^2}{|G|} \cdot |G|^2 = d^2 |G|$. However because of the discretization procedure, it is easier to upper bound by $d^2 |G|^2$ since the cardinal of the discretized version of $\mathcal{A}_{\mathcal{R}}$ is not easily computable. Moreover, the reader will note that the cardinal of the basis has no real significance by itself because the basis was computed with an arbitrary discretization procedure, and therefore another procedure may have lead to another cardinal. Due to the artifacts during the discretization procedure the basis we construct $\mathcal{B}_{i \to i+1}$ and only approximate a subset of all equivariant maps. We now compute $\|\|\mathcal{B}_{i \to i+1}\|\|$ when employing the $\| \cdot \|_\infty$ on each feature space. Applying the triangular inequality we get:

$$\|\|\mathcal{B}_{i \to i+1}\|\| \leq |\mathcal{B}_{i \to i+1}| \max_{b_{i \to i+1,k} \in \mathcal{B}_{i \to i+1}} \|\|b_{i \to i+1,k}\|\|.$$

It remains then to upper-bound $\|\|b_{i \to i+1,k}\|\|$ for every element in the basis. For all $x \in \mathcal{A}_{\mathcal{R}}$ and for all $p, q \in [|G|]$ denote $b_{i \to i+1,p,q,x}$ the convolution with the equivariant kernel $\mathbf{K}_{G,x}^{p,q}$. We have using a result from da Cunha et al. (2022b) that $\|\|b_{i \to i+1,p,q,x}\|\| \leq \|\mathbf{K}_{G,x}^{p,q}\|_1$. Then, in a non-discretized kernel setting, while noticing that the orbit of $x$ has $|G|$ elements, one has $\|\mathbf{K}_{G,x}^{i,j}\|_1 \leq |G|$. Then, $\|\|b_{i \to i+1,p,q,x}\|\| \leq |G|$. For the identity this remains true as by using of circulant matrices it is trivial that $\|\mathbf{K}_{G,0}^{i,j}\|_1 = |G|$.

## F    ADDITIONAL MATERIAL ON THE PERMUTATION EQUIVARIANT NETWORKS

### F.1    PROOF OF SLT ON PERMUTATION EQUIVARIANT NETWORKS

The aim of this appendix is to prove Corollary 2. The building blocks of the layers are here $\mathbb{F}_i = \mathbb{R}^{n^{k_i}}$. Taking direct sums of them we obtain $\mathbb{F}_i^{n_i} = \mathbb{R}^{n^{k_i} \times n_i}$. Again as in appendix section E.4, the pointwise ReLU is equivariant and furthermore we facilitate the computation of $\|\|\mathcal{B}_{i \to i+1}\|\|$ by working with $\| \cdot \|_\infty$, which implies that the ReLU is 1-Lipschitz. As explained above, the norm that we must consider on $\mathbb{F}_i^{n_i} = \mathbb{R}^{n^{k_i} \times n_i}$ to apply theorem 1 is the max of the norm across the blocks, i.e. still $\| \cdot \|_\infty$ on $\mathbb{R}^{n^{k_i} \times n_i}$. First, observe that $\|\|\mathcal{B}_{i \to i+1}\|\| = n^{k_i} + 1$.

*Proof.* One can check that the worse case scenario happens when making $\sum_{b_k \in \mathcal{B}_{i \to i+1}} b_k$ act on a tensor $\mathbf{X} \in \mathbb{R}^{n^{k_i}}$ full of 1. Denote by $\mathbf{Y}_a$ for $a \in [n]^{k_i}$ the tensor in $\mathbb{R}^{n^{k_i}}$ such that it has a 1 at the index $a$ and 0 everywhere else. The tensor full of 1 is therefore $\sum_{a \in [n]^{k_i}} \mathbf{Y}_a$

$$\|\mathcal{B}_{i\to i+1}\| = \max_{\|\alpha\|_\infty \le 1} \max_{\|\mathbf{X}\|_\infty \le 1} \left\|\left(\sum_k \alpha_k b_k\right) \mathbf{X}\right\|_\infty$$

$$= \left\|\left(\sum_{b_k \in \mathcal{B}_{i\to i+1}} b_k\right)\left(\sum_{a \in [n]^{k_i}} \mathbf{Y}_a\right)\right\|_\infty$$

$$\le \left\|\left(\mathbb{I} + \sum_{\mu \in [n]^{k_i+k_{i+1}}/\mathcal{Q}} B^\mu\right)\left(\sum_{a \in [n]^{k_i}} \mathbf{Y}_a\right)\right\|_\infty$$

$$\le \max_{b \in [n]^{k_{i+1}}} \left|\left(\left(\mathbb{I} + \sum_{\mu \in [n]^{k_i+k_{i+1}}/\mathcal{Q}} B^\mu\right)\left(\sum_{a \in [n]^{k_i}} \mathbf{Y}_a\right)\right)_b\right|.$$

Now, $\forall b \in [n]^{k_{i+1}}$,

$$\left(\left(\mathbb{I} + \sum_{\mu \in [n]^{k_i+k_{i+1}}/\mathcal{Q}} B^\mu\right)\left(\sum_{a \in [n]^{k_i}} \mathbf{Y}_a\right)\right)_b = 1 + \left(\sum_{a \in [n]^{k_i}}\left(\sum_{\mu \in [n]^{k_i+k_{i+1}}/\mathcal{Q}} B^\mu\right)\mathbf{Y}_a\right)_b$$

$$= 1 + \left(\sum_{a \in [n]^{k_i}} B^{(a,b)}\mathbf{Y}_a\right)_b$$

$$= 1 + \sum_{a \in [n]^{k_i}} 1$$

$$= 1 + n^{k_i}.$$

$\square$

Moreover $|\mathcal{B}_{i\to i+1}| = \tilde{b}(k_i + k_{i+1})$ by definition of the Bell numbers. In fact the interested reader will check that one has $|\mathcal{B}_{i\to i+1}| \le \tilde{b}(k_i+k_{i+1})$ and that the equality happens as soon as $n \ge k_i+k_{i+1}$ (for example with $n = 1$ one can not have an independent vector family of $\tilde{b}(k_i + k_{i+1})$ vectors in $\mathcal{L}(\mathbb{R}, \mathbb{R})$ which is of dimension 1. The argument expressed in Maron et al. (2019) needs $n \ge k_i+k_{i+1}$ to ensure that all the equivalence classes $\mu$ have at least one element. Finally, all the conditions to apply theorem 1 are true and one only need to replace $\max(|\mathcal{B}_{i\to i+1}|, \|\mathcal{B}_{i\to i+1}\|)$ by $\max(\tilde{b}(k_i + k_{i+1}), n^{k_i} + 1)$.

## G   EXPERIMENTAL DETAILS

The purpose of our experiments is to empirically validate the our theory found in the main text, and as a result show that by solving appropriate SUBSET-SUM problems one can prune an overparameterized random network to a target one. In this section of the appendix, we describe the network architectures we use for our experiments, the overparameterization scheme we select in order to be compatible with our claims, and the linear program we solve for each target weight in order to find the sparsification mask which leads to the approximation of the target network by the overparameterized one.

For both MPGNN and E(2)-CNN experiments, we first train a single target network on the supervised tasks that we described in table 3. The architecture we use for each of the target networks is described in the tables 4, 5, and 6 below. Notice that we do not utilize bias in the parameterized layers, as well as we do not make use of learnable element-wise affine transformations in the batch normalization layers. We train for 50 epochs using AdamW as the optimizer with learning rate 0.015 and default momentum parameters $\beta = (0.9, 0.999)$ and a cosine scheduler. The weight decay coefficient is set to $5e-4$. For the transductive learning tasks on Cora and CiteSeer with the MPGNN, we define an epoch as 10 parameter updates. For the image classification tasks on RotMNIST and FlipRotMNIST

with the E(2)-CNN the batch size is set to $64$. Finally, the target model is selected as the one which achieves the best validation accuracy throughout training.

Afterwards, we define the overparameterized network. In particular, for each parameterized layer (linear or equivariant) of the target network we declare a module that we are going to approximate it with. The module consists of the composition of three layers; the first and the last being of the same type as the target layer, and the middle one is an element-wise ReLU activation function. We make source that the shapes of the input and output tensors match. We initialize these modules using iid drawn samples from $\mathcal{U}([-a, a])$, where $a$ is determined as twice the maximum absolute parameter of the target network. As we explain in the appendix section A, this is compatible with our theorem.

For each parameter in the target network we solve two SUBSET-SUM problems, one to approximate the positive input tensors and one to approximate the negative input tensors. This distinction is needed if we want to use a ReLU in the overparameterized layers. The width is overparameterized by multiplying the input tensor size with a number that scales proportionally to a hyperparameter constant factor $C$, and logarithmically in the input and output size, the number of layers to be approximated, and in $1/\epsilon$, where $\epsilon$ is the desired network approximation error. For our experiments, we use $\epsilon = 1e-2$. For further details, the reader is requested to examine the associated Python repository that we provide.

Finally, we solve each defined SUBSET-SUM problem by treating it as a mixed-integer linear program, similar to Pensia et al. (2020). Each one of the problems amounts to a different constraint optimization problem of the following form:

$$\min_{z \in \mathbb{R}, \boldsymbol{m} \in \mathbb{Z}_2^n} z \tag{28}$$
$$\text{s.t.} \quad y - \boldsymbol{m}^\top \boldsymbol{x} <= z$$
$$\boldsymbol{m}^\top \boldsymbol{x} - y <= z$$

In the optimization problem above, $\boldsymbol{x}$ is a vector resulting from the multiplication of the two weight matrices which participate in the diamond-shaped approximation scheme for each target weight $y$, as explained in 1. Optimization variables $z$ and $\boldsymbol{m}$ amount to the absolute weight approximation error and part of the binary mask of the second layer in the overparameterized network, which is responsible for approximating the particular weight $y$.

| Layer | $\rho_{\text{in}}$ | $\rho_{\text{out}}$ |
|---|---|---|
| GCNConv | data features | 16 |
| ReLU & Dropout(prob. = 0.5) | | |
| GCNConv | 16 | 16 |
| ReLU & Dropout(prob. = 0.5) | | |
| GCNConv | 16 | number of classes |

Table 4: Target network architecture for the MPGNN experiments.

| Layer | $\rho_{\text{in}}$ | $\rho_{\text{out}}$ |
|---|---|---|
| R2Conv(kernel = (9, 9)) | number of channels $\times$ trivial repr. | 24 regular repr. |
| InnerBatchNorm | | |
| Pointwise Max Pool(kernel = (2, 2)) | | |
| Pointwise ReLU | | |
| R2Conv(kernel=(7, 7)) | 24 regular repr. | 48 regular repr. |
| InnerBatchNorm | | |
| Pointwise ReLU | | |
| GroupPooling | | |
| Max Pool(kernel = (2, 2)) & Flatten | | |
| Linear | $48 \times |G|$ | 48 |
| BatchNorm & ReLU | | |
| Linear | 48 | number of classes |

Table 5: Target network architecture for the E2CNN experiments.

| Layer | $\rho_{\text{in}}$ | $\rho_{\text{out}}$ |
|---|---|---|
| Linear $2 \mapsto 2$ | data features $\times N^2$ | $16 \times N^2$ |
| Linear $2 \mapsto 1$ | $16 \times N^2$ | $32 \times N$ |
| Sum Pooling across $N$ & Flatten | | |
| Linear | $32$ | $64$ |
| ReLU | | |
| Linear | $64$ | number of classes |
| Linear $2 \mapsto 2$ | data features $\times N^2$ | $24 \times N^2$ |
| ReLU | | |
| Linear $2 \mapsto 2$ | $24 \times N^2$ | $48 \times N^2$ |
| ReLU | | |
| Linear $2 \mapsto 1$ | $48 \times N^2$ | $96 \times N$ |
| Sum Pooling across $N$ & Flatten | | |
| Linear | $96$ | $96$ |
| ReLU | | |
| Linear | $96$ | number of classes |

Table 6: Target network architecture for the $k$-order $\mathcal{S}_n$-equivariant GNN experiments. Top architecture was used for the Proteins dataset, whereas the bottom one for the NCI1 dataset.

