# OpenReview forum: "A General Framework For Proving The Equivariant Strong Lottery Ticket Hypothesis"
_ICLR.cc/2023/Conference — ICLR 2023 poster_

### Official Review · Reviewer_Tcgp · 2022-10-25

**Confidence:** 1
**Clarity, Quality, Novelty And Reproducibility:** None
**Correctness:** 3
**Technical Novelty And Significance:** 3
**Empirical Novelty And Significance:** 3
**Recommendation:** 6

**Strength And Weaknesses:**

### Strength:

1. This work proposes a theoretical justification of the strong lottery ticket hypothesis for general equivariant networks.
2. This paper is well-organized, starts from the theoretical results of general equivariant networks, and follows by specific case studies as well as empirical evidence.
3. Experiments are conducted across several equivariant networks.

### Weakness:

1. Whether the proposed framework can further be extended to equivariant networks with other activation functions besides point-wise ReLU?
2. Minor: The y-axis is covered by other subfigures in Figure 3.

**Summary Of The Paper:**

This work introduces a unifying framework to prove the strong lottery ticket hypothesis for general equivariant networks. Firstly, the authors prove that any fixed width and depth G-equivariant network that uses a point-wise ReLU activation function can be approximated with high probability to a pre-specified tolerance by pruning a randomly initialized overparameterized G-equivariant network to a G-equivariant subnetwork. Additionally, this work proves such a prescribed overparametrization scheme is optimal and provides a corresponding lower bound. Experiments on E(2)-steerable CNNs, k-order GNNs, and MPGNNs validate the proposed theory.

**Summary Of The Review:**

None

---

> ### Author Response · Authors · 2022-11-13
> **Rebuttal**
>
> We thank the reviewer for their thoughtful comments and feedback on our work. We are especially appreciative that the reviewer views our work as providing a strong “theoretical justification” for the strong lottery ticket hypothesis for general equivariant networks. We are also heartened that the reviewer feels our paper is well-organized and that our experiments are conducted across several equivariant models. We now answer the key points raised by the reviewer:
>
> **Extension to activation functions besides point-wise ReLU**
>
> We appreciate the reviewer's comment regarding the extension of our theory to other activation functions beyond the pointwise ReLU. As we highlight in our response to Reviewer g7iD a recent work by Burkholz 2022 [1] extends the strong lottery ticket setting for dense networks with ReLU activations to dense networks whose pointwise nonlinearity have derivatives at the right and the left at $0$ and whose sum is non-zero. The class of activation functions that satisfy these properties include well-known ones such as arctan, sigmoid, tanh, and all real $C^1$ functions whose derivative is non-zero at $0$.  We believe a similar proof strategy used in the dense nets setting can be leveraged in our use cases but we leave this extension as a fruitful direction for future work.
>
> **Fixing Figure 3**
>
> We thank the reviewer for pointing this out to us. We have updated the figure to improve the readability of the y-axis.
>
> **References**
>
> [1] Burkholz, R. (2022). Most Activation Functions Can Win the Lottery Without Excessive Depth. arXiv preprint arXiv:2205.02321.

---

### Official Review · Reviewer_THqA · 2022-10-25

**Confidence:** 4
**Correctness:** 4
**Technical Novelty And Significance:** 2
**Empirical Novelty And Significance:** 2
**Recommendation:** 8

**Clarity, Quality, Novelty And Reproducibility:**

**Clarity, Quality, Novelty:**
- The paper is fairly well written with minor typos and formatting errors as mentioned above.
- I think the notation is somewhat dense, and while I feel it can be made simpler, it is not obvious to me how.
- As I mentioned above, the proofs and statements seem extremely similar to Pensia et al., so I would hope that there is an easier way to transfer those results to this domain. The authors cite Pensia et al., but I would have liked them to acknowledge the extent to which they borrowed techniques from there.

**Strength And Weaknesses:**

**Strengths:**
1. The paper seems to be technically sound and a generalization of several known Strong LTH results. I am not yet convinced that Equivariance is an important property of DNNs but it seems that there is some interest in the area.
2. The paper is fairly well written with few typos which I will point out.

**Weaknesses:** (I will also include my questions for clarification here)
1. In the end of section 2 it is stated "With no assumption on the inputs, ..." - I don't follow this statement. Can you elaborate?
    - Also, merely a few paragraphs later, the inputs are assumed to be bounded in the unit $\ell_2$ sphere and the same decomposition is invoked. So I don't follow the criticism of Pensia et al. here.

2. Minor typos:
    - In Section 3.1, "To approximates $f$..." is a typo.
    - In Section 3.2, "is far from optimal as their result give guarantees.." is a typo.
    - In Section 4.2 and a few other places, the $\times$ operator seems to be typeset incorrectly. Please check.
    - Please use \citep{} when citing multiple works in the end of a sentence.

3. End of Section 3.1 states that this result improves on the overparameterization of Pensia et al., since G-equivariant networks have fewer effective parameters.
    - However, Pensia et al., prove a result about pruning an overparameterized dense network to recover a dense target network while the result here is to approximate  G-equiariant net. Does this result give a tighter bound even when approximate dense networks? Otherwise, the comparison is incorrect and unfair.

4. Section 3.2 states that the lwoer bound is (tight)
    - This is untrue. The bound is loose by a $\log(n)$ factor which they claim can be culled since it arose from a poor choice norm. However, in the appendix they say that a rescaling "might" eliminate the term.
    - It is also stated that by choosing the weights carefully, it can be eliminated. But the randomness is essential for the definition of a Strong LT. Also, if one is allowed to choose the weights exactly, you could choose a bespoke overparamterization for each target network which makes the entire result useless.
    - I'm still not convinced that the $\log(n)$ term is easy to get rid of.

5. In Section 1, it is stated that "Moreover, Theorem 1 can be extended to the settings where overparamatrized networks have depth $L+1$ as in Burkholz (2022b)"
    - However, all the results in the paper seemed to need $2L$ depth. Can you point me where this is shown?

6. Table 3 results:
    - The overparameterization required in the results seems to be extremely high. The experiments are in-line with Pensia et al., but the overparameterization there seemed to be more like $10$. Also, in Ramanujan et al., the required overparmaterization is $2$ or $4$. Is there a reason the overparamterization is so large here?

7. Lemma 3 in Appendix:
    - It is stated that these results are extensions of those specified in Pensia et al. I believe this result is stated as a remark in Pensia et al. Please check.

8. Appendix B.3: Lower bound:
    - I don't understand several of the statements following Theorem 2. It is stated that the assumptions are "only used to ensure that the lower bound holds" - I'm not sure how this is a valid justification of the assumption. It seems fairly essential to the proof.
    - I agree that the assumptions are mild and have no problem with their usage, but these statements are somewhat misleading.

9. Table 4 and Table 5 are very hard to read. Please clean them up.

10. Similarity to Pensia et al.:
    - The proof techniques, statements and even the lower bound seem almost identical to Pensia et al. with a switch in terminology to account for equivariant networks.
    - While I don't question that this is a valid contribution, I can't help but feel there is a simpler way to transfer these proofs to this domain.

**Summary Of The Paper:**

The paper provides a framework for proving the Strong Lottery Tickey Hypothesis (SLTH) for G-equivariant networks based on the proof by Pensia et al. They show that within this framework, several of the known results (Pensia et al., da Cunha et al., Burkholz et al.) can be considered instantiations of the general theorem.

**Summary Of The Review:**

I think the paper is a reasonable contribution. I have some questions and clarifications but if those are answered, I would recommend acceptance.

---

> ### Author Response · Authors · 2022-11-13
> **Rebuttal Part 1/3**
>
> We would like to thank the reviewer for their time and valuable feedback when reviewing our paper. We appreciate their constructive criticisms that aided us in formulating our responses which we used towards updating our draft of our paper. We now address the key questions below:
>
> **Q1. Assumptions on the input**
>
> We appreciate the reviewer's concern regarding the role of input assumptions with the respect to the seminal work of Pensia et. al 2020 [1].
> We believe there is a potential source for confusion in our original phrasing which was not intended to refer to Pensia et. al 2020[1]’s assumptions. Our statement "With no assumption on the inputs, ..." was in reference to the additional assumption from da Cunha et. al 2022 [2] where all inputs to the CNN are positive. Our proof techniques and in particular Theorem 1 does not require such positivity assumptions.
> We have updated our manuscript to make clear that our assumption on bounded inputs in a unit ball is the same as the assumption done by Pensia et. al 2020 and that our improvement is toward da Cunha et. al 2022 [2]'s assumptions.
>
> **Q2. Minor Typos**
>
> We thank the Reviewer for catching these typos. We have fixed them in the updated draft.
>
> **Q3. Overpametrization of Dense Networks vs. Equivariant Networks**
>
> We understand the reviewers healthy skepticism regarding to the level of overparametrization. We first clarify that the reviewer is indeed correct in understanding that when Theorem 1 is applied to the same setting as Pensia et. al 2020 [1] there is no improvement—i.e. $G$ is the trivial group and we prune dense networks to approximate dense networks. However, as soon as the group $G$ is a non trivial finite group which is the main setting of this paper we do see improvements in the overparametrization factor needed in comparison to Pensia et. al whose pruning strategies would result in a $G$-equivariant network to be pruned as if it was a normal dense network. It is critical to re-emphasize that our pruning strategy preserves equivariance which results in a pruned $G$-equivariant network that has a smaller number of effective parameters than a dense network of the same size. We can explicitly compute the gain in number of effective parameters in the prunable overparametrized network—i.e. its effective size is decreased by a factor: $ \frac{n_i n_{i+1}}{|B_{i \to i+1}|}$
> (in the case of a two layer network with $n_i$, $n_{i+1}$ the input and output dimensions and $|B_{i \to i+1}|$ the dimension of the space of equivariant maps between the input and output space).
> We believe the factor that we gain by pruning an equivariant network rather than a dense network is a key contribution of our work as we maintain the desirable equivariance property while still having a lower overparametrization footprint. We acknowledge that our original phrasing of the statement at the end of section 3.1 may have been unclear but we hope that this answer satisfies the reviewer and clears up the chief source of confusion.
>
> Part 1/3

---

> > ### Author Response · Authors · 2022-11-13
> > **Rebuttal 2/3**
> >
> > **Q4. Tightness of Lowerbound**
> >
> > We thank the reviewer for pointing out a subtle discussion regarding the term $\log(n)$ in the bound in Theorem 1. First, we want to emphasize that this is only an informal discussion rather than a rigorous argument and whose chief aim is to show the conditions of the term $\log(n)$ which may seem not natural and to give provide some hints to get eliminate it.
> > The $\log(n)$ is more precisely $\log(n_in_{i+1}\max(|B_{i \to i+1}|, |||B_{i \to i+1}|||))$.
> >
> > To start off, the term $n_in_{i+1}|||B_{i \to i+1}|||$ arises from our choice of how we want to approximate the target network: we want to bound the error on the output by an error $\epsilon$ for inputs in the unit ball. Therefore we need to approximate each weight of the target network by an error of $\frac{\epsilon}{n_in_{i+1}}$. What we meant by “a change of norm” was only to change the definition of approximating a network (i.e. the “norm” in the space of networks: approximating a network by $\epsilon$ would mean approximating each weight of the target network by $\epsilon$. This is of course a different setting than the one in Pensia et al and in our Theorem 1 but that allows us to remove the term $n_in_{i+1}||||B_{i \to i+1}|||$.
> >
> > Secondly, we explain that the term $n_in_{i+1}|B_{i \to i+1}|$ arises from the probabilistic setting of the Strong Lottery Ticket Hypothesis (SLTH). Since we want to approximate ALL the weights with probability $1-\delta$, and since there are basically $n_in_{i+1}|B_{i \to i+1}|$ different weights to approximate, we must take a probability $\frac{\delta}{n_in_{i+1}|B_{i \to i+1}|}$ in Lemma 2 and the factor $n_in_{i+1}|B_{i \to i+1}|$ appears in the bounds. We can get rid of it by changing the setting of the SLTH: the weights are no longer drawn from a random process but from a deterministic initialization such that all possible target networks can be achieved by pruning the deterministic but smartly initialized overparametrized network with probability one! We emphasize here that the initialization of the overparametrized network does not depend on the target network which is unknown at this stage. Such an initialization can be taken for example by decomposing the overparametrized network in the different blocks of the diamond shape and taking the weights in each block to be $\pm 1, \pm ½, \pm ¼, \pm (½)^{\log(\epsilon) / \log(2)} = 1/\epsilon$. You can approximate each weight of the target network by pruning the diamond shape with a mask which is the binary writing of the target weight.
> >
> > To conclude, these two ideas are possible avenues to annihilate the term $\log(n)$ from the bounds but this requires a change of setting, and the reviewer is right here to be critical of our informal discussion as this was not sufficiently clear. We have now clarified these points in the updated draft.
> >
> > **Q5. Extension of Theorem 1 to depth $L+1$ as in Burkholz 2022b**
> >
> > We appreciate the reviewer's comment regarding the extension of our theory to networks of depth $L+1$. While we believe that such a result is indeed possible we stated this as a remark in our manuscript and did not provide a formal proof. We have clarified that in the manuscript. One avenue for obtaining such a result is to consider the proof technique used in Burkholz 2022 [3] and extend it to equivariant networks of depth $L+1$ in a similar manner as we extended the proof techniques of Pensia et. al 2020 for equivariant networks of depth $2L$ (i.e. pruning in an equivariant basis). However, we note such an endeavor is beyond the scope of this current work due to space considerations. Furthermore, we believe following such a strategy may lead to bigger bounds with respect to the size of the target network in terms of the number of parameters of the overparametrized network. We agree with the reviewer that the current discussion of this is misplaced in the introduction of the paper. As a result, we have shifted these statements to section 6 (Discussion) and clearly demarcated this as a possible direction for future investigation.
> >
> > Part 2/3

---

> > > ### Author Response · Authors · 2022-11-13
> > > **Part 3/3**
> > >
> > > **Q6. Overparametrization factor in comparison to Pensia et. al**
> > >
> > > The Reviewer makes an astute observation regarding the overparametrization factor reported in our work versus the one used in Pensia et. al 2020 [1]. We clarify this source of confusion by first noting that the number we report corresponds to a different quantity than the one that Pensia et al. 2020. Specifically, in table 3, we report the total number of parameters in the overparameterized network divided by the number of parameters in the target network. Please note that the total number of parameters in the overparameterized network includes the pre-pruned neurons which are set to zero in order to make the diamond-shaped approximation scheme arise from the composition of two fully-connected layers. In contrast to us, Pensia et al., section 5, the number reported is the size of the sets used to perform subset-sum algorithm which is needed to approximate each weight.
> > > Our choice of overparametrization is entirely motivated by the manner we present our theoretical results (e.g. Theorem 1). In particular, in our implementation the corresponding number is about $\sim 100$, however we correctly use $2 \cdot C \cdot \log(\frac{K}{\epsilon})$ as demanded by theorem 1, where $K \propto l \cdot d^2$ and the constant factor $2$ is due to us by-passing of the ReLU non-linearity. In contrast, the approximate number reported by Pensia et al. in section 5 refers to $C \cdot log(\frac{1}{\epsilon})$. As a result, we argue that if Pensia et al 2020 reported the exact overparametrization factor in a similar manner as we do the per-weight factor between the two sets of numbers would be on the same scale and numerically close to one another. We understand that this technicality was not sufficiently clear in the original draft of the manuscript but we hope our response here completely addresses the reviewer's original concern.
> > >
> > > **Q7. Clarifications regarding Lemma 3**
> > >
> > > It was indeed stated as a remark but was not proved and since we wanted the paper to be self-contented we added this as a lemma. We cited Pensia moreover just before the lemma. We will add nevertheless that it was stated as a remark.
> > >
> > > **Q8. Appendix B.3: Lower bound**
> > >
> > > We appreciate the reviewer's concern regarding the lower bound in Theorem 2. We would first like to clarify that there are two results in the theorem. For the lower bound on $\Theta$ we only need the first assumption in appendix B.3 which is completely necessary. Such an assumption is present under different forms in Da Cunha et. al 2022 [2] and Pensia et. al 2020 [1]. The final two assumptions in B.3 are very mild and are useful in proving only the second part of the theorem which gives a lower bound on $\tilde{n}_i$. Our previous phrasing  “This assumption is only used to achieve the lower bound on the overparametrization factor in the theorem, i.e. $\tilde{n}_i \geq \Omega(n_i \log\left(\frac{1}{\epsilon}\right))$.” was specifically in reference to this point. We understand that this sentence may lead to confusions and we have clarified this in the updated draft. Finally, if the reviewer has further questions regarding this point we would be happy to clarify further
> > >
> > > **Q9. Table 4 and Table 5 are very hard to read.**
> > >
> > > We thank the reviewer for pointing this out. We have updated the manuscript.
> > >
> > > **Q10. The similarity of proof techniques to Pensia et. al**
> > >
> > > We acknowledge the reviewer's concern regarding the proof techniques we borrow in relation to Pensia et. al and how they could be perceived as similar. We would like to first highlight we have responded to a similar concern to Reviewer 3WVk.
> > >
> > > We would first like to gently push back against this assertion as Pensia et. al tackle dense networks wherein there are no convolution operations. In contrast, Da Cunha et. al 2022 show that in CNNs the use of convolution requires a more fine-grained analysis and treatment to achieve the same type of SLT result, moreover they require all inputs to be positive. While the spirit of our work is built on top of foundations laid by Pensia et. al and Da Cunha et. al we argue that it is not at all an obvious insight that we must overparametrize and prune in an $G$-equivariant basis. As a result of this constraint, we are unable to directly apply Pensia et al. 20's techniques to bypass the ReLU non-linearities, nor Da Cunha et. al’s technique for that matter. We detail the challenges of the analysis in Section 3 "Challenges in Adapting Proof Techniques" of our manuscript.
> > >
> > > Part 3/3

---

> > > > ### Author Response · Authors · 2022-11-13
> > > > **Rebuttal: Thank you**
> > > >
> > > > We would like to thank the reviewer for their review of our paper. We believe we have answered all the great points raised by the reviewer in our author response. We respectfully ask the reviewer to reconsider their impression of the paper and potentially improve the given score if the raised concerns have been allayed. We thank the reviewer again for their time and we are also happy to answer any further questions that arise!
> > > >
> > > > **References**
> > > >
> > > > [1] Pensia, A., Rajput, S., Nagle, A., Vishwakarma, H., & Papailiopoulos, D. (2020). Optimal lottery tickets via subset sum: Logarithmic over-parameterization is sufficient. Advances in Neural Information Processing Systems, 33, 2599-2610.
> > > >
> > > > [2] da Cunha, A., Natale, E., & Viennot, L. (2022, April). Proving the strong lottery ticket hypothesis for convolutional neural networks. In International Conference on Learning Representations.
> > > >
> > > > [3] Burkholz, R. (2022). Most Activation Functions Can Win the Lottery Without Excessive Depth. arXiv preprint arXiv:2205.02321.

---

> > > > > ### Comment · Reviewer_THqA · 2022-11-16
> > > > > **Response to authors**
> > > > >
> > > > > Thank you for your detailed responses! You have indeed addressed almost all of my concerns. The remark about the construction using the binary expansion seems closely related to Sreenivasan et al. "Finding Nearly Everything within Random Binary Networks" where they prove the strong LTH for binary random networks.
> > > > >
> > > > > I am still slightly skeptical about the importance of equivariant networks, but this is a good contribution to the strong lottery ticket literature. I will be increasing my score and recommending acceptance.

---

> > > > > > ### Author Response · Authors · 2022-11-17
> > > > > > **Thanks for the reference!**
> > > > > >
> > > > > > We are delighted to hear that the reviewer is satisfied with our answers in the rebuttal and we heartily thank them for increasing their score!
> > > > > >
> > > > > > Our construction of using the binary expansion is indeed very closely related to the one used in Sreenivasan et. al 2022 [4] since they both use the binary decomposition of a real number to approximate a fixed weight of a target network by a subnetwork of an overparametrized one. The major difference is that our construction uses a network of depth $2$ to approximate the target weight as opposed to Sreenivasan et. al 2022 who use a deeper net but with only binary weights. Furthermore, our usage of this construction differs from Sreenivasan et. al 2022 as we seek to demonstrate that smart—but deterministic—initialization of an overparametrized network can lead to all possible target networks being recovered via pruning the overparametrized one (with pruning masks that depend on the target network). In contrast, the goal in Sreenivasan et. al 2022 is to show that it is possible to approximate any target network by pruning a polylogarithmically overparametrized random binary network. We find this use case both interesting and compelling.
> > > > > >
> > > > > > We thank the reviewer again for pointing out the similarity between our construction and the one used in Sreenivasan et. al 2022 and we have updated our paper to include a citation of this work. We are happy to answer any further questions the reviewer might have.
> > > > > >
> > > > > > [4] Sreenivasan, K., Rajput, S., Sohn, J. Y., & Papailiopoulos, D. (2022, May). Finding Nearly Everything within Random Binary Networks. In International Conference on Artificial Intelligence and Statistics (pp. 3531-3541). PMLR.

---

### Official Review · Reviewer_3WVk · 2022-10-26

**Confidence:** 4
**Correctness:** 4
**Technical Novelty And Significance:** 3
**Empirical Novelty And Significance:** 3
**Recommendation:** 6

**Clarity, Quality, Novelty And Reproducibility:**

* The clarity and organization of the writing is very good.
* The proofs seem correct.
* Novelty: see above.

**Strength And Weaknesses:**

Strengths:
* The SLTH is of great interest to the community, and this papers’ results seem to nicely generalize all previous papers, in particular recovering the result for CNN.
* The construction is neat and easy to follow.

Weaknesses:
* The setting of SLTH is not novel, and has been previously studied. The proof techniques in this paper are also quite similar to the proof techniques of previous papers, but generalized to equivariant networks in general (beyond MLPs and CNNs). Nevertheless, I understand that this is not an actionable criticism for the authors.

**Summary Of The Paper:**

This paper proves the Strong Lottery Ticket Hypothesis (SLTH) in the context of equivariant neural networks. The SLTH states that for every network f, and every sufficiently overparametrized, randomly-initialized network h, there is a pruning of h that recovers f.

Prior work: In the context of fully-connected networks, the SLTH was proved by Malach et al.’20 and then strengthened by Pensia et al.’20, who gave upper and lower bounds.
The SLTH was extended to CNNs by da Cunha et al.’22 and Burkholz ’22, where the key new technical contribution was to give a pruning scheme that preserved translation equivariance of the network.

This work: This work generalizes the arguments of Pensia et al’20 and Burkholz’22 to apply to any equivariant networks. The pruning scheme bears similarity to Pensia et al.’20, but with the main difference that it works by pruning weights in an equivariant basis.
The main theorem proving SLTH is discussed in the contexts of E(2)-steerable networks and permutation equivariant networks, and experiments are given to verify its validity.


**Summary Of The Review:**

This paper is well-written and unifies and generalizes previous results on Strong Lottery Ticket Hypothesis for MLPs and CNNs. I recommend acceptance.

---

> ### Author Response · Authors · 2022-11-13
> **Rebuttal**
>
> We thank the reviewer for their feedback. We appreciate their nuanced comments. We are glad that the reviewer found our contribution "easy to follow". We put a lot of effort into breaking down to key elements of our contribution, we are thus glad it worked. We now
> address Reviewer 3WVk's main concern that regards the novelty of the setting and of the proof techniques.
>
> **The setting of SLTH is not novel … but generalized to equivariant networks in general (beyond MLPs and CNNs).**
> We appreciate the reviewers concern with respect to the novelty of our framework to study equivariant SLTs.  We now address these concerns in the following two points:
> 1. We believe extending the question of SLTH to non-dense architecture is a highly nontrivial question. Particularly, in our contribution, we ask the architecture of the target networks to be the one of an equivariant network. This slight change in the target and source architecture drastically changes the problem and raises the following new questions:
>   - Can we prune equivariant architectures to approximate equivariant neural networks? We answer yes to this question in Thm 1.
>   - If the target network is equivariant, is it more efficient to use a source architecture that is equivariant? We also answer yes to this question in Thm 1 and 2.
>
> A standard dense neural network could be used as a source architecture to approximate an equivariant target. But, as discussed in the paragraph after Thm 2, the overparametrization needed would be significantly larger with a dense net than with an equivariant net.
>
> 2. Regarding the proof techniques. One of the main difficulties is that we cannot prune arbitrarily the source network because by doing so we would lose its equivariance properties. Thus, the constraints we have on how we can prune the source network prevent us from directly applying Pensia et al. 20's techniques to bypass the ReLU non-linearities. We detail the emerging challenges of the analysis in Section 3 "Challenges in Adapting Proof Techniques".
>
>
> We would like to thank the reviewer for their time in reviewing our paper. We believe we have answered all the great points raised by the reviewer in our author response and we respectfully ask the reviewer to reconsider their impression of the paper and potentially improve the given score if the raised concerns have been sufficiently well addressed. We are also happy to answer any further questions that the reviewer might have. Please do let us know!

---

### Official Review · Reviewer_g7iD · 2022-10-28

**Confidence:** 4
**Correctness:** 4
**Technical Novelty And Significance:** 3
**Empirical Novelty And Significance:** 4
**Recommendation:** 8

**Clarity, Quality, Novelty And Reproducibility:**

- The paper is clear and easy to read, and has a very clear motivation.
- The quality of the results, in my opinion, is solid. These results are extensions of existing results, with a change in the pruning strategy, but they do provide a unified perspective on them, which is useful. To the best of my knowledge no such unifying perspective on SLTH for G-equivariant networks exists.
- Have the authors considered just working with the Clebsch-Gordan products? (Such as in Clebsch-Gordan Nets?) They also provide a way to simply work with a quadratic non-linearity in Fourier space, avoiding pointwise non-linearities altogether.

**Strength And Weaknesses:**

- The problem is well-motivated and important. G-equivariant networks are now becoming increasingly important in applications, especially in the physical sciences. It is thus important to understand if an equivalent of the strong lottery ticket hypothesis also holds for them.
- The results are solid to the best of my knowledge (although I have only worked through the proofs while skipping some details). The approach is fairly obvious -- they adapt earlier work, but not to prune the weights themselves, but the coefficients that are used to combine the equivariant bases.
- The case-studies for the 4 cases considered are useful to position the results in context.
- The results generalize earlier results for MLPs and CNNs.
- Experimental resuls show a proof of concept that their approach is able to find such sub-networks.
- Paper is well-written and is easy to follow.

**Summary Of The Paper:**

Recent work has shown that the strong lottery ticket hypothesis (SLTH) can be extended to classical convolutional neural networks, with a similar level of overparameterization as for dense networks. CNNs are translation equivariant (ignoring pooling and edge effects), so a natural question to ask if the SLTH can be extended to more general G-equivariant networks. The main contribution of this paper is to show SLTH can indeed be generalized to G-equivariant networks.

The question whether there should exist equivariant strong lottery tickets for sufficiently overparatermized G-equivariant NNs is not trivial at first, but in hindisight fairly obvious (as explained below). The lack of triviality partly stems from the fact that the sub-network extraction can easily result in loss of equivariance, defeating the purpose. Nonetheless, the authors show that this is possible i.e. a target GCNN with fixed width and depth (and which uses a pointwise non-linearity like the ReLU) can be approximated with high probability by a subnetwork in a random GCNN which has double the depth, and has the width increased by a log factor. This subsumes results of Pensia et al. and Orseau et al. for MLPs and that of Bukrholz et al. and da Cunha et al. for CNNs. The authors also show that a log factor increase in width is actually optimal, and this work irrespective of the overparameterization strategy employed.

The authors rely on the use of a G-equivariant basis -- which is a basis of equivariant linear maps between two vector spaces. The error metric the authors consider (for the approximation quality of the subnetwork) is uniform approximation over a unit ball. The approaches of Pensia et al. da Cunha et al., Burkholz et al. are adapted to work for the equivariant case. More specifically, the subset sum approach fails if applied as is. The idea for pruning then becomes pruning out the weights that combine the equivariant basis (rather than pruning in a canonical basis as parameters of the weight matrices can't be directly pruned). Thus what might seem non-trivial at first is fairly obvious -- we simply need to prune the parameters that combine elements of this basis. However, this introduces an additional difficulty of dealing with pointwise non-linearities like ReLU effectively -- which is an additional technical challenge that the authors address.

Finally, the authors present case studies for specific choices for G. For the translation group their results generalize earlier results. The main theorem is also applied for the E(2) steerable case, permutation equivariant networks, and message passing networks. Experimental results show that the proposed approach is able to approximate the target network adequately.

**Summary Of The Review:**

See above.

---

> ### Author Response · Authors · 2022-11-13
> **Rebuttal**
>
> We thank the reviewer for their positive appraisal of our manuscript. We are heartened to hear that our problem setting is “well-motivated” and the presented contributions are “solid” and the four case-studies are “useful” to position the results in context. Finally we thank the reviewer for considering our manuscript to be well-written and easy to follow.
>
> **Extension to Clebsch-Gordon Networks**
>
> The reviewer poses an interesting question as to whether our method can be extended to Clebsch-Gordon networks. We were not aware of this line of work but to our understanding, we can view this extension as a particular case of a more general question concerning the application of our method to other activation functions beyond the pointwise-ReLu—Clebsch-Gordon networks being ones that use an equivariant quadratic non-linearity. In the more general case, but still with pointwise nonlinearities, Burkholz 2022 [1] extends Pensia et. al 2020 [2] now seminal result by observing that the domain of usage of the non-linearity is concentrated around small values. Specifically, the extension concerns all pointwise non-linearities which have derivatives on the right and on the left at zero and whose sum is non-zero. Most common functions (especially all $C^1$ functions whose derivative at zero is non zero) have this property. The square function on the other hand—whose derivative at zero is precisely zero—does not satisfy this property. We can still tackle this problem by adding a linear term to the non linearity (that remains equivariant) and then the derivative at zero can be taken to be non zero. Despite this, there remains a second problem which is that the non-linearity does not act pointwise in real space nor in Fourier space. Such a fact does not allow to use the techniques we need for our methodology but it still leaves open for possible technical novelties that can be potentially leveraged. We leave these extensions and their applications as future work.
>
> [1] Burkholz, R. (2022). Most Activation Functions Can Win the Lottery Without Excessive Depth. arXiv preprint arXiv:2205.02321.
>
> [2] Pensia, A., Rajput, S., Nagle, A., Vishwakarma, H., & Papailiopoulos, D. (2020). Optimal lottery tickets via subset sum: Logarithmic over-parameterization is sufficient. Advances in Neural Information Processing Systems, 33, 2599-2610.

---

> > ### Comment · Reviewer_g7iD · 2022-11-17
> > **Thanks for the comment!**
> >
> > Thanks for the thoughtful comment. Indeed, you are right that it might not be trivial to extend it to "fully Fourier" type equivariant networks. I will take a detailed look at the updates and comment if I have any questions.

---

### Decision · Program_Chairs · 2023-01-20

**Decision:**

Accept: poster

**Justification For Why Not Higher Score:**

I believe although this work is interesting, the overall scope may be limited to a small community within the theory of ML.

**Justification For Why Not Lower Score:**

Because the paper is novel enough and contributes to the theory

**Metareview: Summary, Strengths And Weaknesses:**

The strong lottery ticket hypothesis (SLTH) postulates that a randomly initialized neural network can be pruned to obtain a subnetwork that approximates well another target network. This is possible given enough overparameterization. This paper investigates whether the SLTH can be extended to more general G-equivariant networks, which exhibit a certain symmetry in their operations (e.g., this can be thought of as an extension of CNNs). The paper demonstrates that this is indeed possible  and provide experimental results to show the effectiveness of their proposed approach.

The reviewers were quite positive about this work, and listed the following strengths and weakness for the paper

Strengths:

- this work extends the SLTH to an interesting family of models.
- The paper provides a unifying lens on previous results for dense networks and CNNs.
- The experimental results demonstrate the effectiveness of the proposed approach.

Weaknesses:

- Some concerns about the setting and techniques being similar to past work, and comparisons being perhaps unlcear
- Some concern about the assumptions made with regards to the inputs of the network

In their rebuttal, the authors provided clarifying feedback to the reviewers, and for the most part settled all significant points of criticism.

This is a paper that will overall be useful and interesting to the folks working on the theory aspects of the SLTH, and should be accepted for publication


**Note From Pc:**

if the above contains the word "oral" or "spotlight" please see: "oral" presentation means -> notable-top-5% and "spotlight" means -> notable-top-25%. As stated in our emails, we are disassociating presentation type from AC recommendations